# The centrosomal protein 83 (CEP83) regulates human pluripotent stem cell differentiation toward the kidney lineage

**Fatma Mansour**[1,2,3], **Christian Hinze**[1,2,4,5], **Narasimha Swamy Telugu**[4,6], **Jelena Kresoja**[7], **Iman B Shaheed**[3], **Christian Mosimann**[7], **Sebastian Diecke**[4,6], **Kai M Schmidt-Ott**[1,2,5]*

[1]Department of Nephrology and Medical Intensive Care, Charité-Universitätsmedizin Berlin, Berlin, Germany; [2]Molecular and Translational Kidney Research, Max Delbrück Center for Molecular Medicine in the Helmholtz Association, Berlin, Germany; [3]Department of Pathology, Faculty of Veterinary Medicine, Cairo University, Cairo, Egypt; [4]Berlin Institute of Health, Berlin, Germany; [5]Department of Nephrology and Hypertension, Hannover Medical School, Hannover, Germany; [6]Technology Platform Pluripotent Stem Cells, Max Delbrück Center for Molecular Medicine in the Helmholtz Association, Berlin, Germany; [7]Department of Pediatrics, Section of Developmental Biology, University of Colorado School of Medicine, Anschutz Medical Campus, Aurora, United States

*For correspondence:
schmidt-ott.kai@mh-hannover.de

**Competing interest:** The authors declare that no competing interests exist.

**Abstract** During embryonic development, the mesoderm undergoes patterning into diverse lineages including axial, paraxial, and lateral plate mesoderm (LPM). Within the LPM, the so-called intermediate mesoderm (IM) forms kidney and urogenital tract progenitor cells, while the remaining LPM forms cardiovascular, hematopoietic, mesothelial, and additional progenitor cells. The signals that regulate these early lineage decisions are incompletely understood. Here, we found that the centrosomal protein 83 (CEP83), a centriolar component necessary for primary cilia formation and mutated in pediatric kidney disease, influences the differentiation of human-induced pluripotent stem cells (hiPSCs) toward IM. We induced inactivating deletions of *CEP83* in hiPSCs and applied a 7-day in vitro protocol of IM kidney progenitor differentiation, based on timed application of WNT and FGF agonists. We characterized induced mesodermal cell populations using single-cell and bulk transcriptomics and tested their ability to form kidney structures in subsequent organoid culture. While hiPSCs with homozygous *CEP83* inactivation were normal regarding morphology and transcriptome, their induced differentiation into IM progenitor cells was perturbed. Mesodermal cells induced after 7 days of monolayer culture of *CEP83*-deficient hiPCS exhibited absent or elongated primary cilia, displayed decreased expression of critical IM genes (*PAX8*, *EYA1*, *HOXB7*), and an aberrant induction of LPM markers (e.g. *FOXF1*, *FOXF2*, *FENDRR*, *HAND1*, *HAND2*). Upon subsequent organoid culture, wildtype cells differentiated to form kidney tubules and glomerular-like structures, whereas *CEP83*-deficient cells failed to generate kidney cell types, instead upregulating cardiomyocyte, vascular, and more general LPM progenitor markers. Our data suggest that *CEP83* regulates the balance of IM and LPM formation from human pluripotent stem cells, identifying a potential link between centriolar or ciliary function and mesodermal lineage induction.

## Editor's evaluation

This paper describes a novel role of the centrosomal protein CEP83 in mesoderm patterning. Specifically, CEP83 is shown to be required for the formation of the intermediate mesoderm and

kidney progenitor tissue derived from it. In a CEP83 knockout situation in human pluripotent stem cells, a shift to lateral plate mesoderm at the expense of intermediate mesoderm occurs, which is convincingly demonstrated. This work, therefore, presents important new insights into the processes involved in mesodermal lineage induction and the fine-tuning of kidney differentiation.

## Introduction

During mammalian embryonic development, the mesoderm forms axial, paraxial, and lateral plate domains that harbor precursor cells for distinct organ systems. Forming as a major part of the lateral plate mesoderm (LPM), the intermediate mesoderm (IM) harbors progenitor cells of all kidney epithelial cells (*Davidson et al., 2019*), whereas the remaining LPM contributes progenitors of various cell types, including cells of the cardiovascular system (*Prummel et al., 2020*). The molecular and cellular mechanisms that drive induction of the IM and distinct LPM domains during embryonic development are not fully understood.

The centrosomal protein 83 (CEP83) is a component of distal appendages (DAPs) of centrioles. DAPs are involved in the anchoring of the mother centriole to the cell membrane, an early and critical step in ciliogenesis (*Lo et al., 2019*; *Tanos et al., 2013*; *Yang et al., 2018*; *Kurtulmus et al., 2018*; *Wheway et al., 2015*; *Bowler et al., 2019*; *Failler et al., 2014*; *Shao et al., 2020*; *Mansour et al., 2021*). CEP83 recruits other DAP components to the ciliary base, and loss of CEP83 disrupts ciliogenesis (*Tanos et al., 2013*). In radial glial progenitors, removal of CEP83 disrupts DAP assembly and impairs the anchoring of the centrosome to the apical membrane as well as primary ciliogenesis (*Yang et al., 2018*; *Shao et al., 2020*). Mutations of CEP83 in humans have been associated with infantile nephronophthisis (*Failler et al., 2014*), an early onset kidney disease that results in end-stage renal disease before the age of 3 years (*Hildebrandt, 2004*; *Luo and Tao, 2018*) and additional organ anomalies (*Failler et al., 2014*). To date, how the loss of CEP83 function contributes to aberrant kidney development remains unclear.

Human-induced pluripotent stem cells (hiPSCs) provide useful tools to study molecular mechanisms of cellular differentiation. Protocols for the induction of kidney organoids from iPSC have been successfully developed (*Takahashi et al., 2007*; *Morizane et al., 2015*; *Taguchi et al., 2014*; *Takasato et al., 2015*; *Freedman et al., 2015*; *Kumar et al., 2019*). The protocol by Takasato et al. uses stepwise exposure of iPSC to WNT and FGF agonists in a monolayer culture system for a 7-day period, which results in the induction of cells with a transcriptional phenotype resembling kidney progenitors in the IM (*Takasato et al., 2015*). Transfer of these cells to an organoid culture system followed by another series of WNT and FGF signals results in the differentiation of three-dimensional (3D) kidney organoids composed of different kidney cell types, including glomerular and tubular cells. Genome editing studies have previously been used to study the effects of genetic defects associated with kidney diseases on kidney differentiation in human iPSC systems (*Freedman et al., 2015*; *Tan et al., 2018*; *Boyle et al., 2008*; *Kobayashi et al., 2008*; *Howden et al., 2019*; *Kuraoka et al., 2020*). Here, we studied the effect of an induced knockout of *CEP83* in human iPSCs on kidney organoid differentiation. We uncovered a novel role of CEP83 in determining the balance of IM versus LPM differentiation, implicating a centrosomal protein in early mesodermal lineage decisions.

## Results

### CEP83 is essential for the differentiation of hiPSCs into kidney cells

To investigate the effect of *CEP83* loss on the differentiation of hiPSCs into IM kidney progenitors, we applied CRISPR-Cas9 technology to induce a null mutation in the *CEP83* gene in hiPSCs (*Figure 1A*). Three hiPSCs clones designated *CEP83*$^{-/-}$ (*KO1*, *KO2*, and *KO3*) carried deletions within *CEP83* exon 7, each of which led to an induction of a premature stop codon resulting in a predicted truncated protein (*Figure 1B–D* and *Figure 1—figure supplement 1A*). These clones exhibited a complete loss of CEP83 protein by immunoblotting (*Figure 1E*). Three wildtype clones were derived as controls (*WT1*, *WT2*, and *WT3*). All six clones were morphologically indistinguishable (by brightfield microscopy) and had similar overall gene expression profiles (by bulk RNA-seq and qRT-PCR), including pluripotency and lineage marker expression (*Figure 1—figure supplement 1B, C*, and *Figure 1—figure supplement 2A, B*). In KO clones, the anticipated altered transcripts of CEP83 were detectable

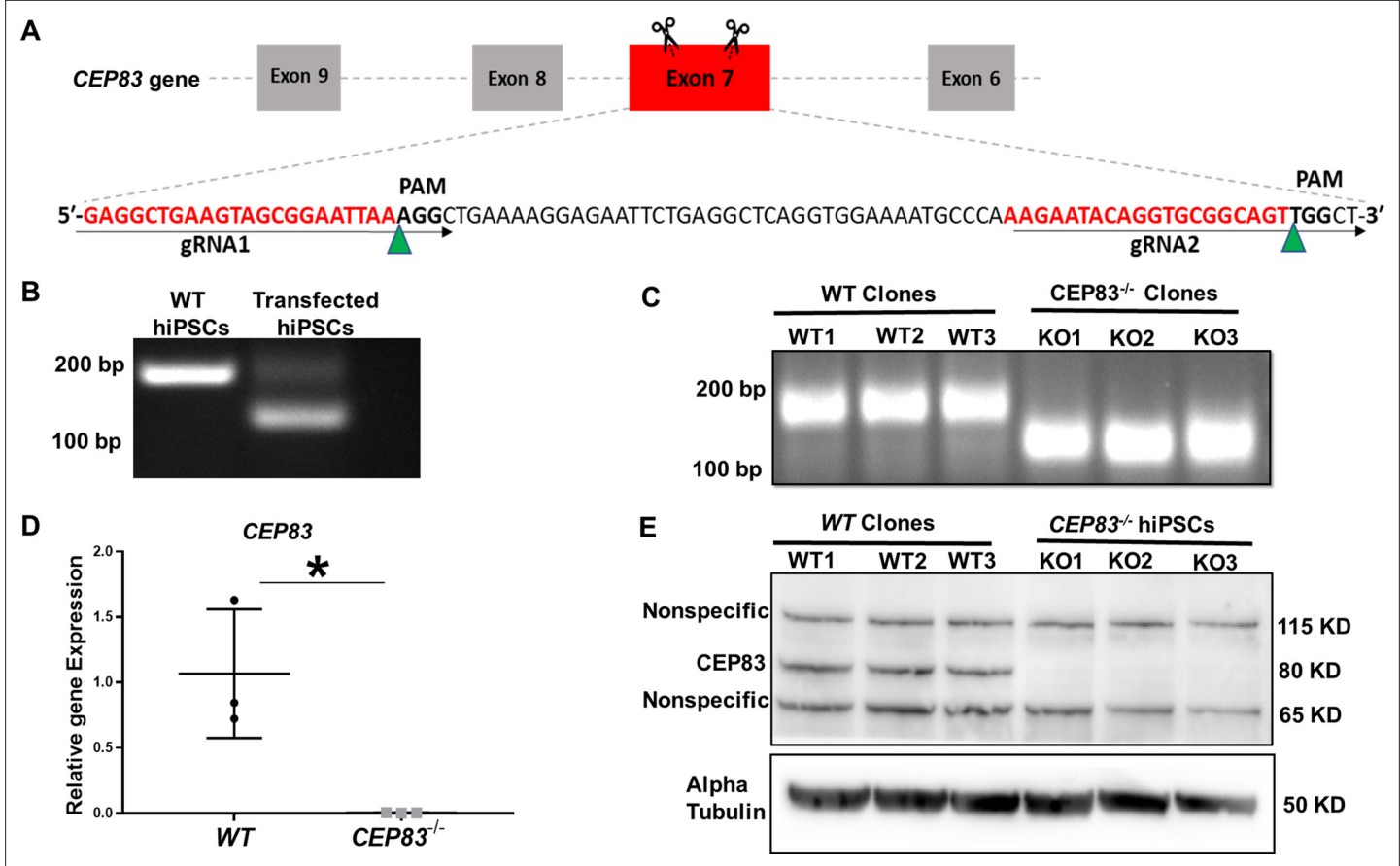

**Figure 1.** Generation of centrosomal protein 83 (CEP83)-deficient human pluripotent stem cells. (**A**) Schematic of the experimental approach to induce a deleting mutation in exon 7 of the *CEP83* gene (as described in the methods section). (**B**) DNA extracted from pooled transfected cells was subjected to PCR, targeting the predicted deletion site in the *CEP83* gene. In addition to the 182 bp fragment present in untransfected wildtype (WT) cells, an approximately 120 bp fragment was detected in transfected cells, corresponding to the induced deletion in exon 7. (**C**) Three clones (*CEP83$^{-/-}$* clones *KO1, KO2, KO3*) carried 62–74 bp deletions within *CEP83* exon 7, which led to an induction of premature stop codons or frameshift mutation on both alleles of *CEP83*. Three wildtype clones (*WT1, WT2,* and *WT3*) were used as controls. (**D**) Quantitative RT-PCR for a fragment corresponding to the deleted region in *CEP83* exon 7 produced a detectable signal in RNA extracts from WT clones but not CEP83$^{-/-}$ clones. (**E**) Immunoblotting of *WT* and *CEP83$^{-/-}$* clones using a CEP83 antibody targeting the C-terminal region of the protein (see Methods for details) indicated a complete loss of the 83 KD band corresponding to CEP83 protein in the three *KO* clones compared with the three *WT* clones. Data are mean ± SD. *p<0.05 and **p<0.01 vs. WT. See *Figure 1—source data 1 and 2*. See also *Figure 1—figure supplements 1–2*.

The online version of this article includes the following source data and figure supplement(s) for figure 1:

**Source data 1.** The file contains detailed original PCR gels and immunoblots.

**Source data 2.** Excel sheet shows RT-qPCR data for mRNA expression of CEP83 in WT and knockout hiPSCs.

**Source data 3.** File contains uncropped PCR gels and immunoblots.

**Figure supplement 1.** *CEP83$^{-/-}$* human-induced pluripotent stem cells (hiPSCs) retain global iPCS cell gene expression signatures and express pluripotency markers.

**Figure supplement 2.** Phenotypical, molecular, and genetic characterization of *CEP83$^{-/-}$* human-induced pluripotent stem cells (hiPSCs) versus the wildtype hiPSCs.

based on bulk RNA-seq (data not shown). Single nucleotide polymorphism - analysis confirmed identical karyotypes of all six clones (*Figure 1—figure supplement 2C*).

Together, these findings confirmed the successful deletion of CEP83 in iPSCs without any overt direct cellular phenotypic consequences. We applied a 7-day monolayer protocol using timed application of WNT and FGF agonists as reported by *Takasato et al., 2015* to differentiate *WT* and *KO* hiPSCs into IM kidney progenitors (*Takahashi et al., 2007*; *Morizane et al., 2015*; *Taguchi et al., 2014*; *Figure 2A*).

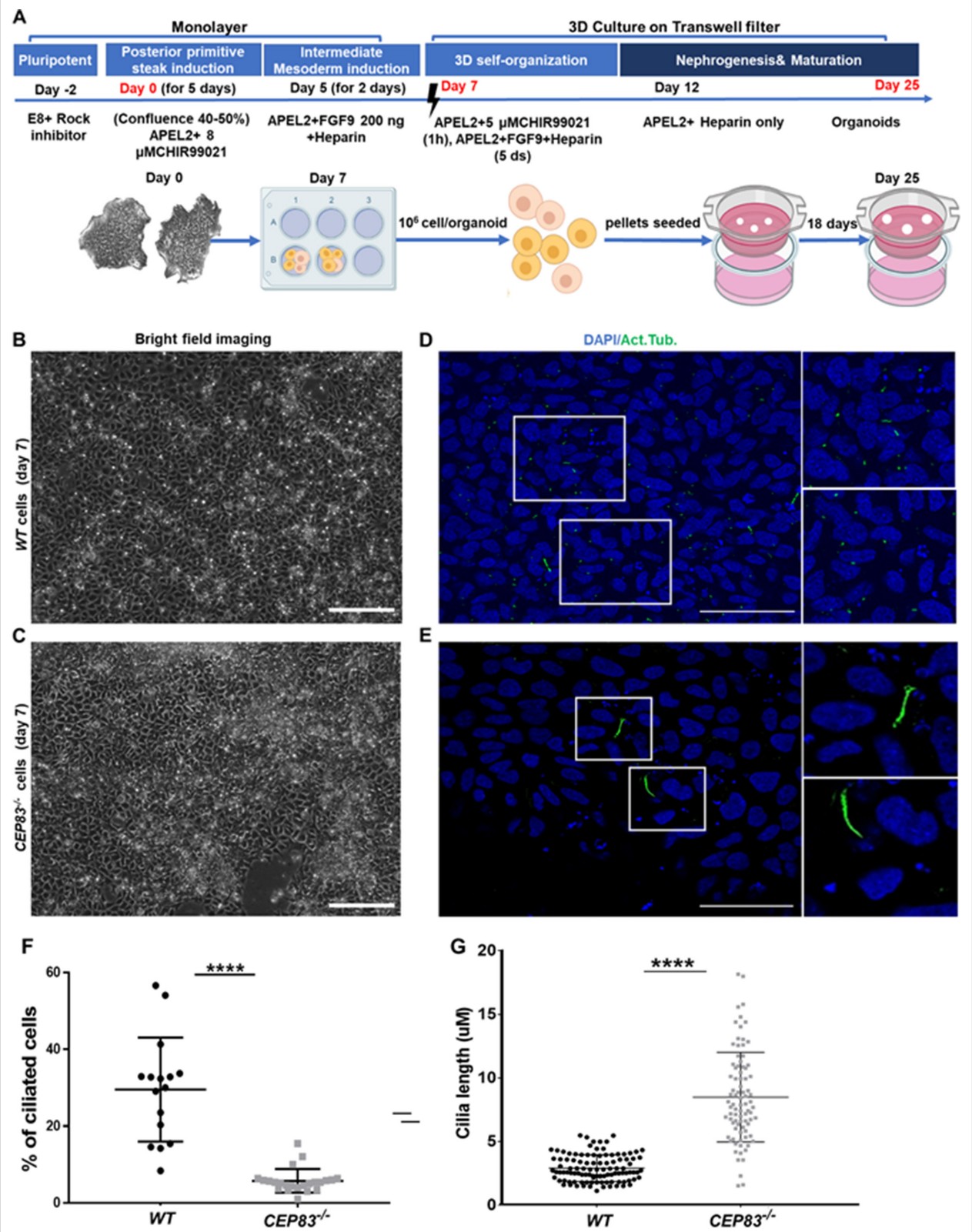

**Figure 2.** Differentiation of CEP83⁻/⁻ human-induced pluripotent stem cells (hiPSCs) to intermediate mesoderm cells (day 7) is associated with defective ciliogenesis. (**A**) The schematic illustrates the applied differentiation protocol of hiPSCs, as previously described by *Takasato et al., 2015*. (**B–C**) *WT* and *CEP83⁻/⁻* cells on 7 days of culture (D7) of differentiation did not show overt morphological differences by brighfield microscopy. (**D–E**) Representative images of *WT* and *CEP83⁻/⁻* cells on D7, immunostained for acetylated tubulin (green) and nuclei (DAPI, blue), revealing fewer and elongated cilia in

*Figure 2 continued on next page*

Figure 2 continued

CEP83⁻/⁻ cells. (F) Quantitative analysis of the percentage of ciliated cells in *WT* and *CEP83*⁻/⁻ cells (D7). (G) Quantitative analysis of the ciliary length in *WT* and *CEP83*⁻/⁻ cells (**D7**). n=3 clones per group. ****p<0.0001. Bar = 50 µm. See *Figure 2—figure supplement 1*.

The online version of this article includes the following figure supplement(s) for figure 2:

**Figure supplement 1.** Loss of *CEP83* in organoids results in defective ciliogenesis.

After 7 days of culture (D7), *WT* and *KO* cells exhibited an indistinguishable morphology by bright field microscopy (*Figure 2B and C*). Immunostaining for acetylated tubulin, however, indicated abnormal primary cilia formation in *CEP83*-deficient cells (*Figure 2D and E*). The number of ciliated cells was reduced from approximately 30% (in *WT* clones) to less than 10% (in *KO* clones) (*Figure 2F*). Among ciliated cells, the length of cilia was increased from 2 to 5 µm (in *WT* clones) to 5–13 µm (in *KO* clones) (*Figure 2G*). This indicated that *CEP83*⁻/⁻ hiPSCs differentiated toward IM progenitors exhibited ciliary abnormalities. To analyze the induced IM kidney progenitor cells functionally, we collected D7 *WT* and *CEP83*⁻/⁻ cells and placed them into an organoid culture system again applying timed WNT and FGF agonists to foster differentiation of mature kidney cell types, as previously reported (*Takasato et al., 2015*; *Figure 2A*). Organoids harvested from *WT* clones after a total of 25 days of culture (D25) had formed patterned kidney epithelial-like structures, including NPHS1-positive glomerulus-like structures, *Lotus tetragonolobus* lectin-positive proximal tubule-like, and E-cadherin (CDH1)-positive distal tubule-like structures (*Figure 3A, C and E*, and *Figure 3—figure supplement 1*). In contrast, *CEP83*⁻/⁻ organoids at day 25 were composed of monomorphic cells with a mesenchyme-like appearance, which stained negative for an array of kidney cell markers (*Figure 3B, D and F*).

Kidney epithelial-like structures formed only in *WT* but not in *CEP83*⁻/⁻ organoids (*Figure 3G*). Similar to the findings in day 7 cells reported above, primary cilia were found in fewer cells of *CEP83*⁻/⁻ organoids (<5% of cells) and were abnormally elongated (*Figure 2—figure supplement 1*).

Next, bulk RNA sequencing of *WT* (WT1, WT2, WT3) and *CEP83*⁻/⁻ (KO1, KO2, KO3) organoids was carried out to evaluate differential gene expression on a genome-wide level, and RT-PCR was used to validate selected genes. Hierarchical clustering of the samples indicated strong gene expression differences between *WT* and *CEP83*⁻/⁻ samples (*Figure 3—figure supplement 2*). Several genes associated with kidney development and kidney epithelial differentiation were differentially expressed with high expression in *WT* organoids but showed comparatively low or absent expression in *CEP83*⁻/⁻ organoids: these genes included kidney-specific lineage genes (*PAX2*, *PAX8*), and lineage/differentiation markers of glomerular cells (*NPHS1*, *PODXL*, *WT1*, *PTPRO*), proximal (*HNF1B*, *LRP2*, *CUBN*), and distal (*EMX2*, *MAL2*, *EPCAM*, GATA3) kidney epithelial cells (*Figure 3H–L*, *Figure 3—figure supplement 2*). This indicated that *CEP83*⁻/⁻ IM progenitors failed to differentiate into kidney cells, suggesting that *CEP83* function is necessary to complete essential steps in the process of differentiation from pluripotent stem cells to kidney cells.

## CEP83 deficiency associates with molecular defects of nephron progenitor cells

We next aimed to gain molecular insights into the lineage impact of *CEP83* deficiency during the course of kidney epithelial differentiation. Since no global transcriptomic differences were detectable between *WT* and *CEP83*⁻/⁻ hiPSCs prior to differentiation (see above), we focused on mesodermal cell stages induced at D7, which displayed mild overall gene expression differences between *WT*- and *CEP83*-deficient cells as detected by bulk RNA sequencing (*Figure 4—figure supplement 1*).

A marked upregulation of nephron progenitor marker genes (*GATA3*, *HOXB7*, *HOXD11*, *EYA1*) (*Bilous et al., 1992*; *Grote et al., 2006*; *Kress et al., 1990*; *Srinivas et al., 1999*; *Wellik et al., 2002*; *Mugford et al., 2008a*; *Ruf et al., 2004*; *Sajithlal et al., 2005*) was observed in both *WT* and *CEP83*⁻/⁻ cells at day 7 (*Figure 4—figure supplement 2*), suggesting that the differentiation path of pluripotent *CEP83*⁻/⁻ cells to IM nephron progenitors was largely intact. To understand the potential molecular defects at the IM stage in more detail, we performed single-cell RNA (scRNA) sequencing on D7 *WT* and *CEP83*⁻/⁻ cells (representing two different hiPSC clones for each condition differentiated in two separate experiments). We obtained transcriptomes from 27,328 cells, representing clones *WT1* (experiment 1: 3768 cells), *WT2* (experiment 2: 5793 cells), *KO1* (experiment 1: 8503 cells), and *KO2*

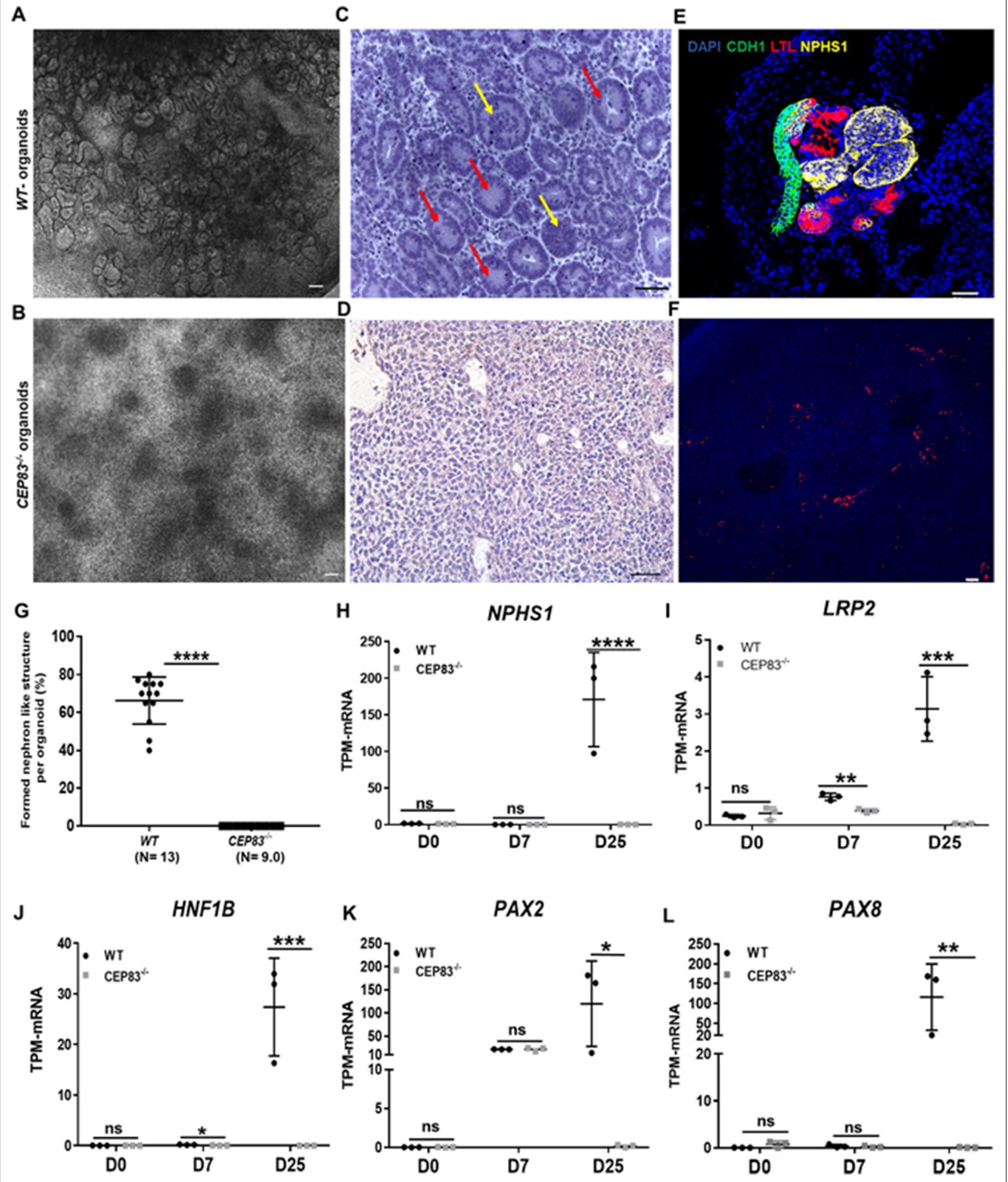

**Figure 3.** Defective kidney organoid differentiation from *CEP83*-deficient pluripotent stem cells. (**A, B**) Brightfield images of organoids after a total of 25 days of culture (**D25**) indicate the formation of multiple kidney-like structures in *WT* organoids (**A**), whereas *CEP83⁻ᐟ⁻* organoids are composed of uniform clusters (**B**). (**C, D**) Representative images of hematoxylin-eosin–stained sections of organoids. *WT* organoids (**C**) display glomerulus-like (yellow arrows) and tubular (red arrow) components, whereas *CEP83⁻ᐟ⁻* organoids (**D**) are composed of monomorphic mesenchymal-like cells.

*Figure 3 continued on next page*

Figure 3 continued

(E–F) Whole mounting immunostaining of organoids for NPHS1 (podocyte marker), LTL (proximal tubule marker), and CDH1 (distal tubule marker) indicates segmented nephron-like structures in *WT* organoids (E), and the absence of such structures in *CEP83⁻/⁻* organoids (F). (G) Quantitative analysis of brightfield images indicating the estimated percentage of organoid area composed of nephron-like structures, organoids were collected from three different experiments. (H–L) Gene expression (transcripts per million [TPM]) of *NPHP1* (H), *LRP2* (I), *HNF1B* (J), *PAX2* (K), and *PAX8* (L) in *WT* and *CEP83⁻/⁻* cells at the indicated time points based on bulk RNA sequencing. n=3 clones per group. Data are mean ± SD. *p<0.05, **p<0.01, ***p<0.001, and ****p<0.0001. ns = not significant. Panels A–F: Bar = 50 μm. See *Figure 3—source data 1 and 2*. See also *Figure 3—figure supplements 1–2*.

The online version of this article includes the following source data and figure supplement(s) for figure 3:

**Source data 1.** The data shows the quantitative analysis of the percent of nephron formation per organoid in knockout organoids versus WT organoids.

**Source data 2.** The sheet shows the plotted TPM values of mRNA sequencing analysis in *Figure 3L-H* for the expression of renal epithelial marker genes in KO and WT at days 0, 7, and 25 of the differentiation.

**Figure supplement 1.** Whole mount immunostaining of the wildtype organoids shows positive staining for NPHS1 (podocyte marker), LTL (proximal tubule marker), and CDH1 (distal tubule marker).

**Figure supplement 2.** mRNA analysis of organoids differentiated for 7+ (18) days indicates marked differences in global gene expression in *CEP83⁻/⁻* (*KO1–KO3*) compared to wildtype (*WT1–WT3*) organoids.

(experiment 2: 9264 cells). Principal component analysis (PCA) using pseudo-bulk expression data of the top 1000 highly variable genes (HVGs) indicated that the first principal component (dimension 1, explaining 54% of expression variation) was driven by the genotype (*WT* vs. *KO*), while the second principal component (dimension 2, explaining 51% of expression variation) was driven by a batch effect of the two experiments (*Figure 4A*). We combined all cells and generated a uniform manifold approximation and projection plot uncovering 10 different cell states/clusters (0–9; *Figure 4B*). We identified marker genes for each cluster (*Figure 4C*), indicating that clusters 1, 3, and 4 represented kidney progenitors/nascent nephrons (expressing, e.g., *PAX8*, *EYA1*, *HOXB7*) in different phases of the cell cycle. Other clusters represented as-of-yet uncharacterized cell types, which were consistent with previous single-cell transcriptome analyses from iPSC-derived cells induced by the same induction protocol (*Subramanian et al., 2019*; *Low et al., 2019*). Each of the four samples (*WT1*, *WT2*, *KO1*, and *KO2*) contributed to each cluster (*Figure 4D*). However, one cluster representing damaged cells (cluster 5) was observed at numerically higher percentages in *KO* cells compared to *WT* cells. Cluster 5 cells expressed high levels of mitochondrial RNAs, and staining for active caspase 3 demonstrated an increased percentage of apoptotic cells in *KO* samples compared to controls (*Figure 4—figure supplement 3*). We focused on kidney progenitors (clusters 1, 3, and 4) and found that a numerically lower percentage of *KO* cells (11.9 and 12.5% in KO clones) contributed to cluster 1 when compared with *WT* cells (25.9 and 36.3% in WT clones) (*Figure 5A*). In contrast, similar percentages of *WT* and *KO* cells were represented in kidney progenitor clusters 3 and 4 (*Figure 5B and C*). Differential gene expression analysis in these three clusters indicated significantly lower expression of kidney progenitor markers *PAX8*, *EYA1*, *CITED1*, and *HOXB7* in *KO* cells from clusters 1, 3, and 4 when compared to *WT* cells (*Figure 5D, E and F*; *Figure 5—figure supplements 1–2*). Interestingly, scRNA sequencing data also showed downregulated expression of genes encoding ciliary proteins, including OFD1, PCM1, and RAB11A (*Ferrante et al., 2006*; *Dammermann and Merdes, 2002*; *Knödler et al., 2010*; *Figure 5—figure supplement 3*), consistent with the ciliogenesis defects in CEP83 knockout cells. These results indicate that *CEP83* deficiency remained permissive with initial kidney progenitor induction, but that these cells exhibited mild molecular defects detectable by differential expression of kidney progenitor genes, which potentially contributed to the failure of *CEP83*-deficient cells to further differentiate toward mature kidney cell types.

## *CEP83* deficiency promotes ectopic induction of lateral plate mesoderm-like cells followed by an expansion of cardiac and vascular progenitors

We next inspected single-cell transcriptomes and bulk RNA sequencing data from D7 cells for genes that were upregulated in *CEP83⁻/⁻* cells compared to WT cells. From this analysis, we observed a consistent upregulation of genes that are normally expressed in early LPM, including *OSR1*, *FOXF1*,

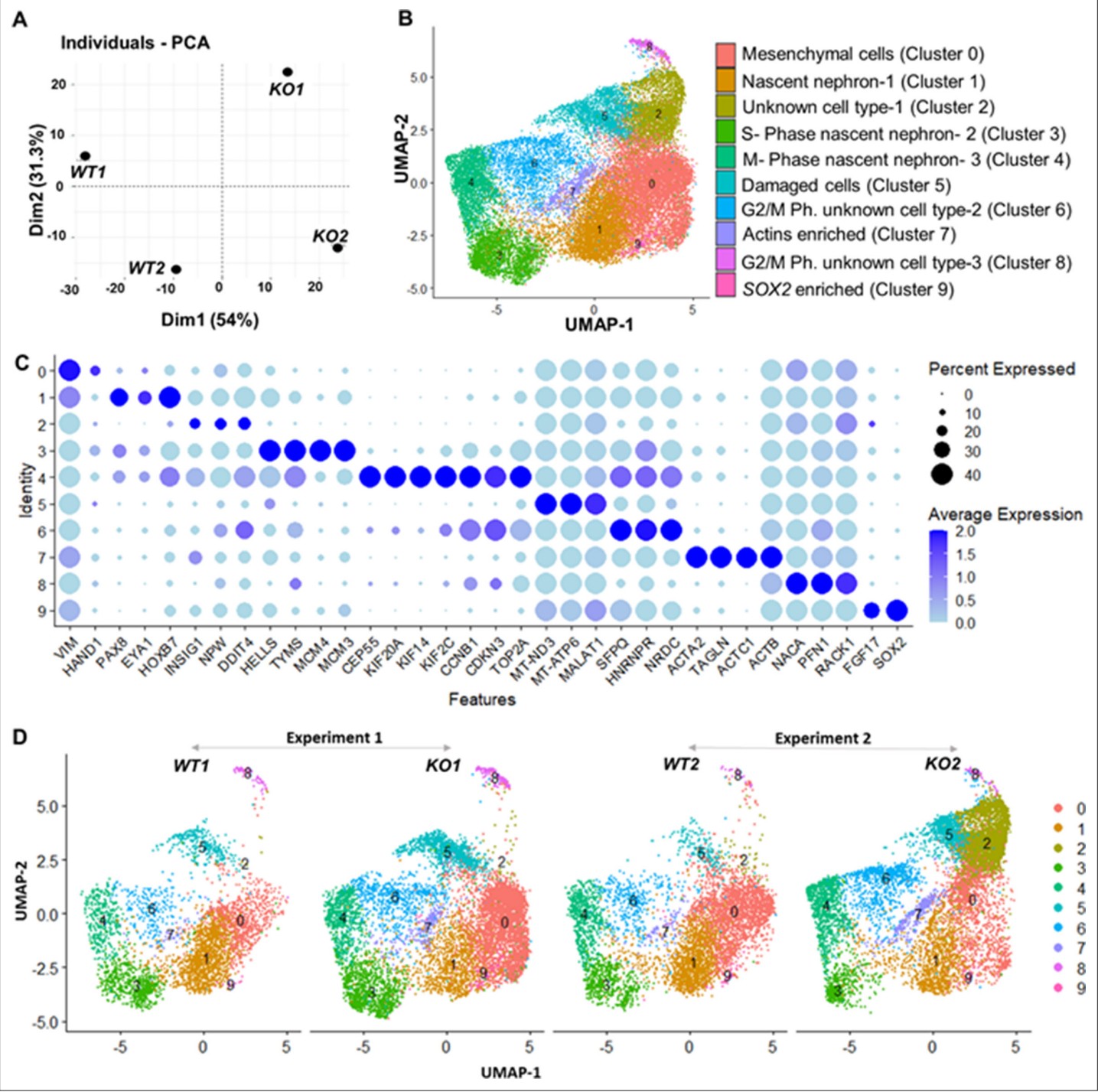

**Figure 4.** Gene expression differences of wildtype (WT) and CEP83⁻/⁻ D7 monolayers based on bulk and single-cell transcriptomics. (**A**) Principal component analysis (PCA) of *WT* (*WT1, WT2*) and *CEP83⁻/⁻* (*KO1, KO2*) cells at day 7 using the average gene expression of the top highly variable 1000 genes in pseudo-bulk scRNA sequencing data. The % variation explained by each PCA axis is indicated in brackets. (**B**) PCA eigenvalues indicate that the principal components, Dim 1 (54%) and Dim 2 (31.3%), account for 85.3% of the expression differences. Dim 1 separates the WT samples from the KO samples, while Dim 2 separates experiment 1 (*WT1, KO1*) from experiment 2 (*WT2, KO2*). (**B**) Uniform manifold approximation and projection (UMAP) of scRNA-seq profiles from 27,328 cells from two wildtype clones (*WT1, WT2*) and two *CEP83⁻/⁻* clones (*KO1, KO2*) derived from two separate experiments (experiment 1: *WT1, KO1*; experiment 2: *WT2, KO2*). Unbiased clustering resulted in 10 clusters, and (**C**) dot plots show expression of selected marker genes of each cluster. (**D**) UMAP plots for *WT* and *KO* samples show the distribution of all clusters per sample (N=2 per group) in B–D. See *Figure 4—figure supplements 1–3*. Source data is available as described in section (Data availability).

The online version of this article includes the following figure supplement(s) for figure 4:

*Figure 4 continued on next page*

*Figure 4 continued*

**Figure supplement 1.** Bulk RNA sequencing shows mild overall gene expression differences between *WT* and *CEP83*-deficient cells at day 7 of differentiation.

**Figure supplement 2.** Expression of intermediate mesoderm marker genes in *WT* and *CEP83*−/− human-induced pluripotent stem cells (hiPSCs) after 7 days of differentiation in a monolayer culture.

**Figure supplement 3.** *CEP83* loss induces apoptosis at day 7 of differentiation.

*FOXF2, FENDRR, HAND1, HAND2, CXCL12, GATA5*, and *GATA6* (***Prummel et al., 2020***; ***Mugford et al., 2008b***; ***Mae et al., 2013***; ***Mahlapuu et al., 2001***; ***Ormestad et al., 2004***; ***Wilm et al., 2004***; ***Wotton et al., 2008***; ***Grote et al., 2013***; ***Schindler et al., 2014***; ***Tsuchihashi et al., 2011***; ***McFadden et al., 2005***; ***Firulli et al., 1998***; ***Risebro et al., 2006***; ***Angelo et al., 2000***; ***Perens et al., 2016***; ***Salcedo and Oppenheim, 2003***; ***Liekens et al., 2010***; ***Loh et al., 2016***; ***Koutsourakis et al., 1999***; ***Holtzinger and Evans, 2007***; ***Reiter et al., 1999***; ***Pikkarainen et al., 2004***; ***Laverriere et al., 1994***; ***Zhao et al., 2008***; ***Prummel et al., 2021***; ***Figure 6A–I***). This suggested that *CEP83*−/− cells entered an aberrant differentiation path assuming a phenotype indicative of broader LPM instead of more specific IM. To further substantiate this idea, we restricted the analysis to progenitor cells of clusters 1, 3, and 4 and to cells from cluster 0, which exhibited a mesenchymal transcriptome fingerprint (see ***Figure 4C***). Within each cell, we analyzed the expression of LPM markers (*FOXF1, HAND1, HAND2*, and *CXCL12*) and of more restricted IM markers (*PAX8, EYA1, and HOXB7*) (***Figure 6—figure supplement 1***). This analysis indicated that *WT* cells of these clusters exhibited an IM-like phenotype, while *KO* cells were shifted toward an LPM-like phenotype. The common IM/LPM marker *OSR1* was expressed at higher level in *KO* cells comparing to the *WT* cells.

We then inspected RNA-seq data from *WT* and *KO* organoids at day 25 for the expression of LPM genes and markers of LPM derivatives. The expression of several LPM genes (*OSR1, FOXF1, FOXF2, FENDRR, HAND1, HAND2,* and *CXCL12*) was strongly upregulated in *KO* cells compared to *WT* cells suggesting that an LPM-like cell pool persisted in D25 *KO* organoids (***Figure 6A–I***). To further substantiate the potential differentiation of the *CEP83*-mutant cells into broadly LPM-like cells, we compared genes that were upregulated genes in D25 organoids (in total, 397 genes) with LPM genes that were previously identified by single-cell transcriptomics of sorted post-gastrulation LPM cells from developing zebrafish (***Prummel et al., 2021***; ***Prummel et al., 2020***; ***Prummel et al., 2022***). Our targeted comparison documented that *CEP83*−/− organoids showed significant enrichment for expression of orthologs of early LPM genes (p=0.006) (***Figure 6—figure supplement 2***), including *OSR1, CXCL12, HAND1/2, KCTD12, PIK3R3*, and *ZBTB2*. A subset of LPM genes enriched for expression in *CEP83*-mutant cells at D25 of differentiation was indicative of cardiac or cardiopharyngeal (*ISL1, TBX1*) as well as vascular progenitor (*SOX7, SOX11, NAP1L3, LMO2, GATA2*) differentiation (***Morikawa and Cserjesi, 2004***; ***Cai et al., 2003***; ***Kwon et al., 2009***; ***Laugwitz et al., 2005***; ***Moretti et al., 2006***; ***Gao et al., 2019***; ***Stennard and Harvey, 2005***; ***Baldini, 2005***; ***Chen et al., 2009***; ***Figure 7A–G***, ***Figure 6—figure supplement 2***). Interestingly, three genes of the upregulated LPM genes, namely OSR1, FOXF1, and FOXF2, are downstream genes of hedgehog signaling (***Han et al., 2017***; ***Kugler et al., 2015***). In addition, HAND2 is upstream of hedgehog signaling (***Anderson et al., 2012***), essential for early embryonic development and regulated by primary cilium (***Kugler et al., 2015***). Furthermore, bulk RNA sequencing data showed significant upregulation of hedgehog signaling components GLI1 and PTCH1 in CEP83−/− cells (***Lee et al., 1997***; ***Villavicencio et al., 2000***; ***Figure 6—figure supplement 3***).

Taken together, these observations document that hiPSCs deficient in *CEP83* respond to an in vitro differentiation program toward kidney progenitors, yet diverge toward a broader LPM progenitor composition without significant IM instead.

## Discussion

This study indicates a novel contribution of CEP83 in regulating the differentiation path from human pluripotent stem cells to kidney progenitors. We pinpoint a stage at day 7 of IM induction where *CEP83* loss of function results in a decreased nephron progenitor pool with downregulation of critical kidney progenitor genes (*PAX8, EYA1, HOXB7*). At the same stage, genes typical of LPM specification (including *FOXF1, FOXF2, FENDRR, HAND1*, and *HAND2*) are upregulated (***Figure 8***). Functionally,

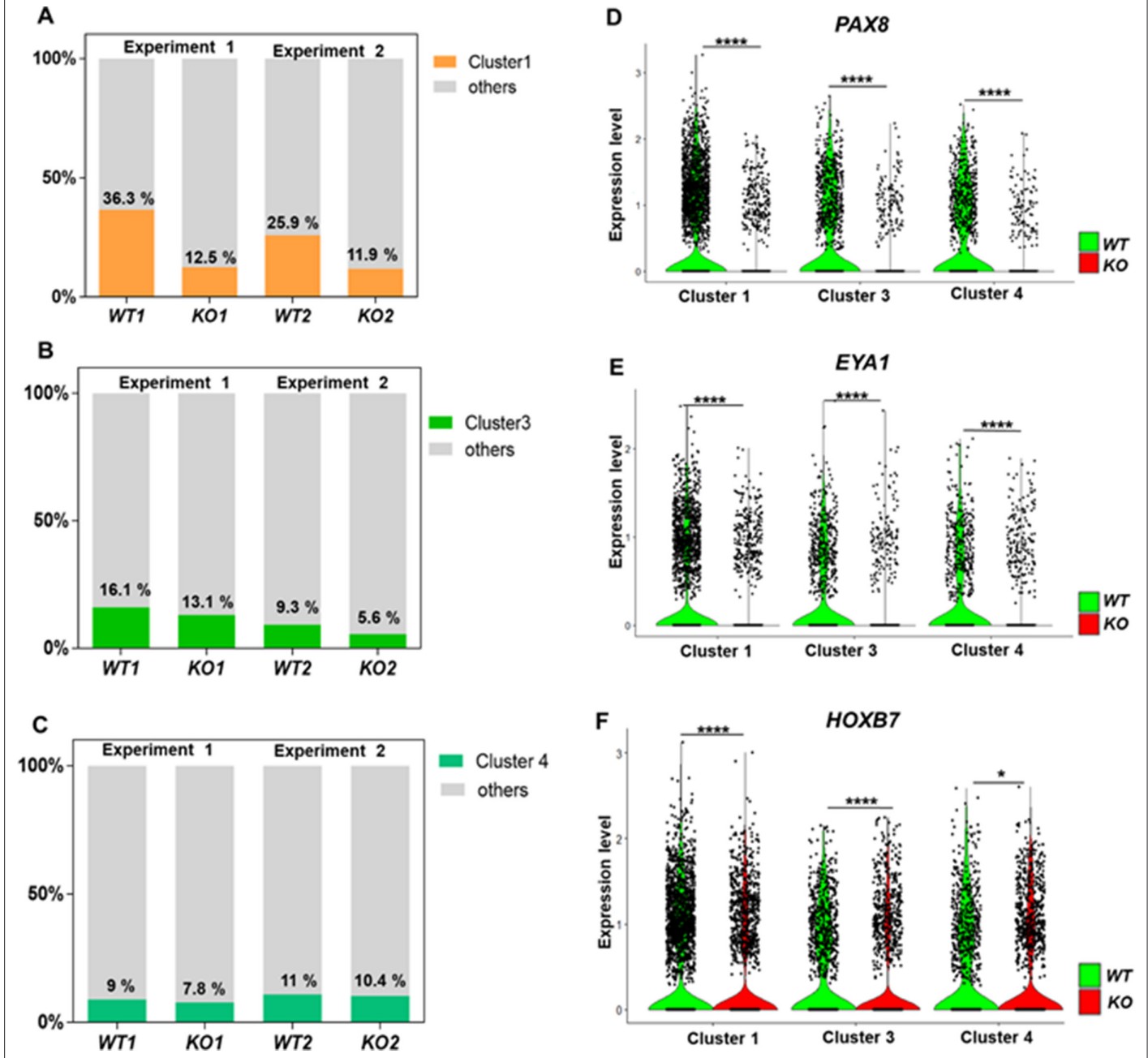

**Figure 5.** Defective kidney progenitor differentiation from *CEP83⁻/⁻* cells after 7 days of monolayer induction. (**A, B, C**) Proportions of cells from kidney progenitor clusters 1 (**A**), 3 (**B**), and 4 (**C**) among *wildtype* (*WT1, WT2*) and *CEP83⁻/⁻* (*KO1, KO2*) cells. (**D, E, F**) Violin plots of gene expression of kidney progenitor genes *PAX8* (**D**), *EYA1* (**E**), and *HOXB7* (**F**) within kidney progenitor clusters 1, 3, and 4 comparing wildtype (*WT*) and *CEP83⁻/⁻* (*KO*) cells. N=2 per group. *p<0.05 and ****p<0.0001. *Figure 5—figure supplements 1–2*. Source data is available as described in section (Data availability).

The online version of this article includes the following figure supplement(s) for figure 5:

**Figure supplement 1.** Expression of selected genes per cluster and per group (*WT* vs. CEP83⁻/⁻).

**Figure supplement 2.** Violin plots of gene expression of kidney progenitor gene *CITED1* within kidney progenitor clusters 1, 3, and 4 comparing wildtype (*WT*) and *CEP83⁻/⁻* (*KO*) cells.

**Figure supplement 3.** Violin plots of single-cell RNA sequencing show downregulated expression of genes encoding ciliary proteins in CEP83⁻/⁻ cells, including (**A**) the basal body protein oral-facial-digital type I *OFD1*, (**B**) pericentriolar material-1 (*PCM1)*, and (**C**) RAS oncogene family 11 A (*RAB11A)*.

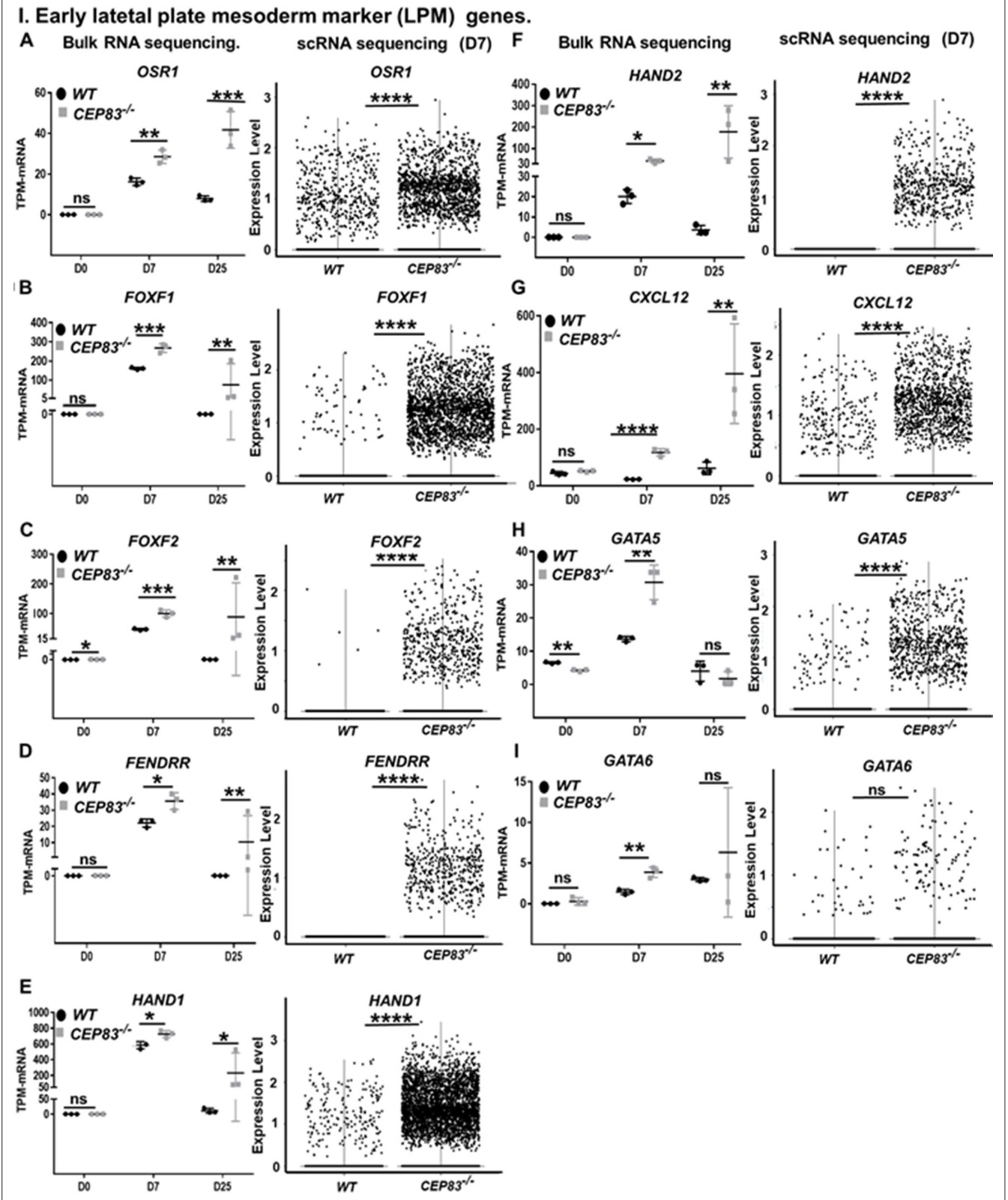

**Figure 6.** *CEP83$^{-/-}$ cells upregulate expression of genes characteristic of early lateral plate mesoderm (LPM). (**A–I**) Expression of early LPM markers OSR1 (**A**), FOXF1 (**B**), FOXF2 (**C**), FENDRR (**D**), HAND1 (**E**), HAND2 (**F**), CXCL12 (**G**), GATA5 (**H**), and GATA6 (**I**) in wildtype (WT) and CEP83$^{-/-}$ cells at day 0 (D0), day 7 (D7), and day 25 (D25) according to bulk RNA-sequencing (left panels) and at D7 according to single-cell RNA (scRNA) sequencing (right panels). N=3 clones per group for bulk RNA seq. N=2 clones per group for scRNA-seq. Expression units are mean transcripts per million (TPM) ± SD.*

*Figure 6 continued on next page*

*Figure 6 continued*

*p<0.05, **p<0.01, ***p<0.001, and ****p<0.0001. ns = not significant. See *Figure 6—figure supplements 1–3*. See also *Figure 6—source data 1*. Check data availability section for other source data.

The online version of this article includes the following source data and figure supplement(s) for figure 6:

**Source data 1.** The sheet shows the plotted TPM values of mRNA sequencing analysis in *Figure 6* for the expression of lateral plate mesoderm marker genes between the KO and WT at days 0, 7, and 25 of the differentiation.

**Figure supplement 1.** *CEP83$^{-/-}$* cells upregulate lateral plate mesoderm (LPM) genes in mesenchymal cells cluster and nascent nephron progenitor clusters.

**Figure supplement 2.** CEP83$^{-/-}$ organoids show significant enrichment compared to developmental zebrafish lateral plate mesoderm (LPM) single-cell RNA (scRNA) data.

**Figure supplement 3.** Upregulation of hedgehog signaling components in CEP83$^{-/-}$ cells.

these alterations are associated with an inability of *CEP83*-deficient cells to form kidney epithelia. Organoids derived from *CEP83*-deficient cells fail to induce any detectable nephron structures, suggesting a novel role for CEP83 during the specification of functional kidney progenitors in the mesoderm.

Our findings are relevant to understanding the cellular and molecular functions of CEP83 and might be relevant to the pathophysiology of human genetic diseases. To date, 11 patients with biallelic mutations of *CEP83* have been reported, 8 of which displayed kidney phenotypes (*Failler et al., 2014*; *Veldman et al., 2021*; *Haer-Wigman et al., 2017*). Available kidney histologies identified microcystic tubular dilatations, tubular atrophy, thickened basement membranes, and interstitial fibrosis.

Extrarenal phenotypes included speech delay, intellectual disability, hydrocephalus, strabismus, retinal degeneration, retinitis pigmentosa, hepatic cytolysis, cholestasis, and portal septal fibrosis with mild thickening of arterial walls and an increase in the number of the biliary canalicules on liver biopsy. Among individuals with *CEP83* mutations, all but one carried at least one missense mutation or short in-frame deletion, suggesting that *CEP83* function may have been partially preserved. One individual with presumed full loss of *CEP83* displayed a more severe phenotype with multiple organ dysfunction. It will be interesting to await future reports of additional *CEP83* mutations in humans and whether complete loss of function alleles will result in broader mesoderm defects or renal agenesis. In this regard, it is interesting that mice with a targeted homozygous loss-of-function mutation of their *CEP83* ortholog (*Cep83$^{tm1.1(KOMP)Vlcg}$*) display midembryonic lethality (at E12.5) with evidence of severe developmental delay as early as E9.5 (https://www.mousephenotype.org/data/genes/MGI:1924298). These phenotypes are potentially consistent with the role of *CEP83* in germ layer patterning and mesoderm development, but a more detailed phenotypical characterization of *Cep83* knockout embryos would be required to substantiate this possibility.

The precise molecular and cellular mechanisms underlying our observations remain to be determined. CEP83 is a protein that is necessary for the assembly of DAPs and primary cilia formation in several cell types (*Tanos et al., 2013*; *Yang et al., 2018*; *Shao et al., 2020*; *Kumar et al., 2021*; *Stinchcombe et al., 2015*; *Joo et al., 2013*). A potential involvement of CEP83-mediated primary cilia formation in the findings reported here is suggested by obvious ciliary defects in CEP83-deficient cells at the D7 and the organoid stage (*Figure 2D–G*, *Figure 2—figure supplement 1*). These defects include reduced percentages of ciliated cells and elongated primary cilia in those cells that continue to form a primary cilium. In addition, CEP83-deficient cells displayed downregulated expression of several transcripts encoding ciliary components (*Figure 5—figure supplement 3*) and evidence of an activation of several hedgehog signaling associated genes. The primary cilium is critically involved in hedgehog signaling (*Ho and Stearns, 2021*). Moreover, Hedgehog signaling is important for mesodermal lineage decisions during gastrulation (*Guzzetta et al., 2020*). This raises the possibility that CEP83 controls mesodermal cell fate decisions by modulating hedgehog signaling in the mesoderm. Nevertheless, additional studies will be necessary to address this possibility. In addition, it remains unknown whether abnormal cilia formation in CEP83-deficient cells causally contributes to the cell fate phenotype.

We observed downregulated expression of the key nephron progenitor genes *PAX8*, *EYA1*, and *HOXB7* in *CEP83$^{-/-}$* cells at day 7, which might explain their failure to differentiate into kidney cells since each of these genes is essential for normal kidney development (*Vincent et al., 1997*; *Pfeffer*

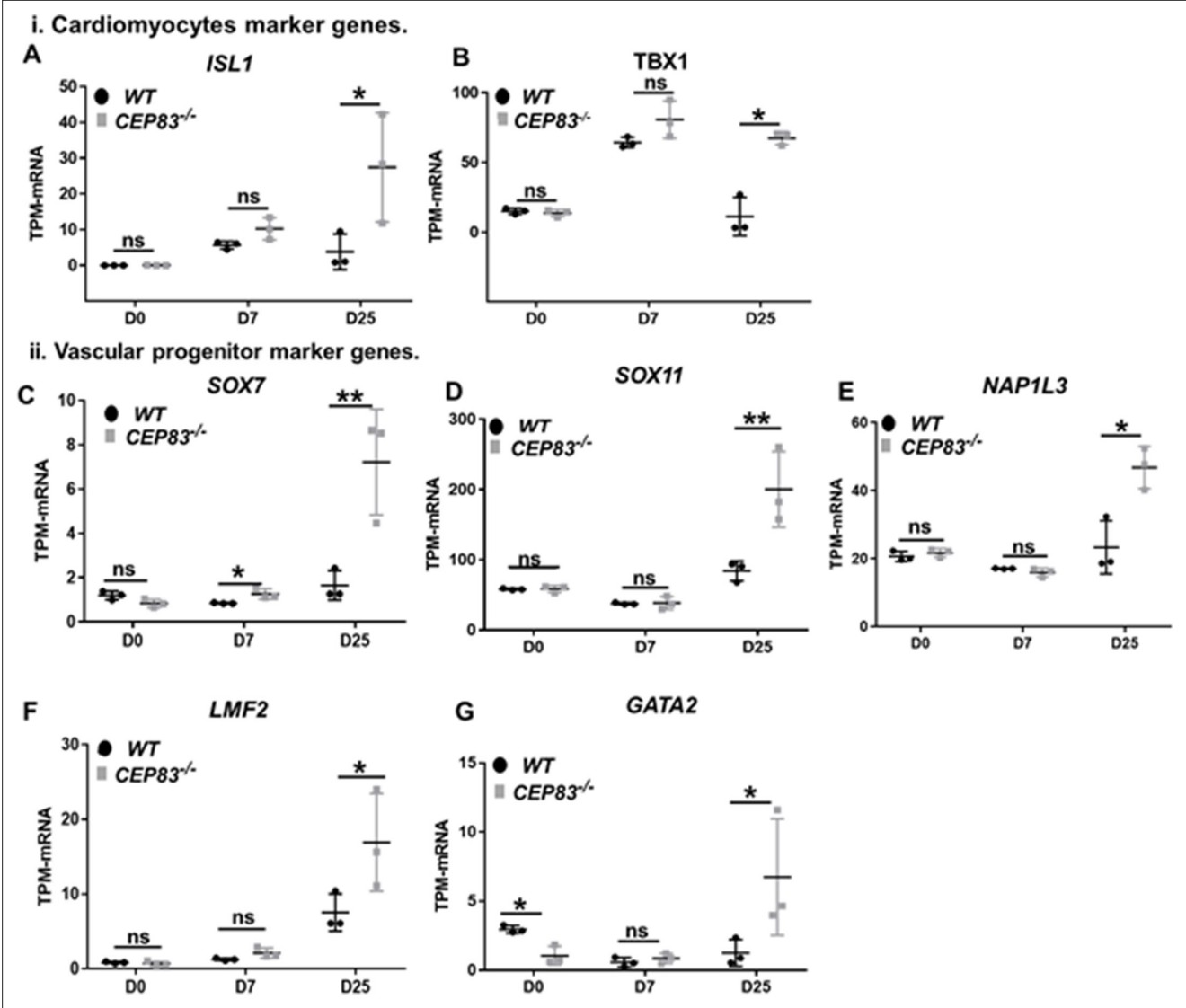

**Figure 7.** *CEP83⁻/⁻* cells upregulate expression of genes characteristic of cardiomyocyte progenitors and vascular progenitors. (**A–G**) Expression of cardiomyocyte markers *ISL1* (**A**), *TBX1* (**B**), and vascular progenitor markers *SOX7* (**C**), *SOX11* (**D**), *NAP1L3* (**E**), *LMO2* (**F**), and *GATA2* (**G**) in wildtype (*WT*) and *CEP83⁻/⁻* cells at day 0 (D0), day 7 (D7), and day 25 (D25) according to bulk RNA-sequencing. N=3 clones per group for bulk RNA seq. Expression units are mean transcripts per million (TPM) ± SD. *p<0.05, and **p<0.01. ns = not significant. See *Figure 6—figure supplement 2*. See also *Figure 7—source data 1*. Check the data availability section for other source data.

The online version of this article includes the following source data for figure 7:

**Source data 1.** The sheet shows the plotted TPM values of mRNA sequencing analysis in *Figure 7* for the expression of cardiomyocytes and vascular progenitors marker genes between the KO and WT at days 0, 7, and 25 of the differentiation.

*et al., 1998*; *Kumar et al., 1998*; *Bouchard et al., 2002*; *Xu et al., 1999*; *Patterson and Potter, 2004*; *Rojek et al., 2006*). Furthermore, inductive signals from HOXB7-positive ureteric bud are known to maintain viability of nephron progenitor cells in the IM (*Barasch et al., 1997*), which may contribute to increased numbers of apoptotic cells we observed in CEP83⁻/⁻ cells. Defects during nephron progenitor differentiation in the IM would be expected to result in severe kidney phenotypes such as renal agenesis or renal hypodysplasia. Defects of centriolar components or cilia have previously been linked to such phenotypes: in mice, centrosome amplification, i.e., the formation of excess centrosomes per cell severely disrupts kidney development, resulting in depletion of renal progenitors and renal hypoplasia (*Dionne et al., 2018*). In humans, loss of KIF14, a protein necessary for proper DAP assembly and cilium formation, has been associated with kidney malformations, including

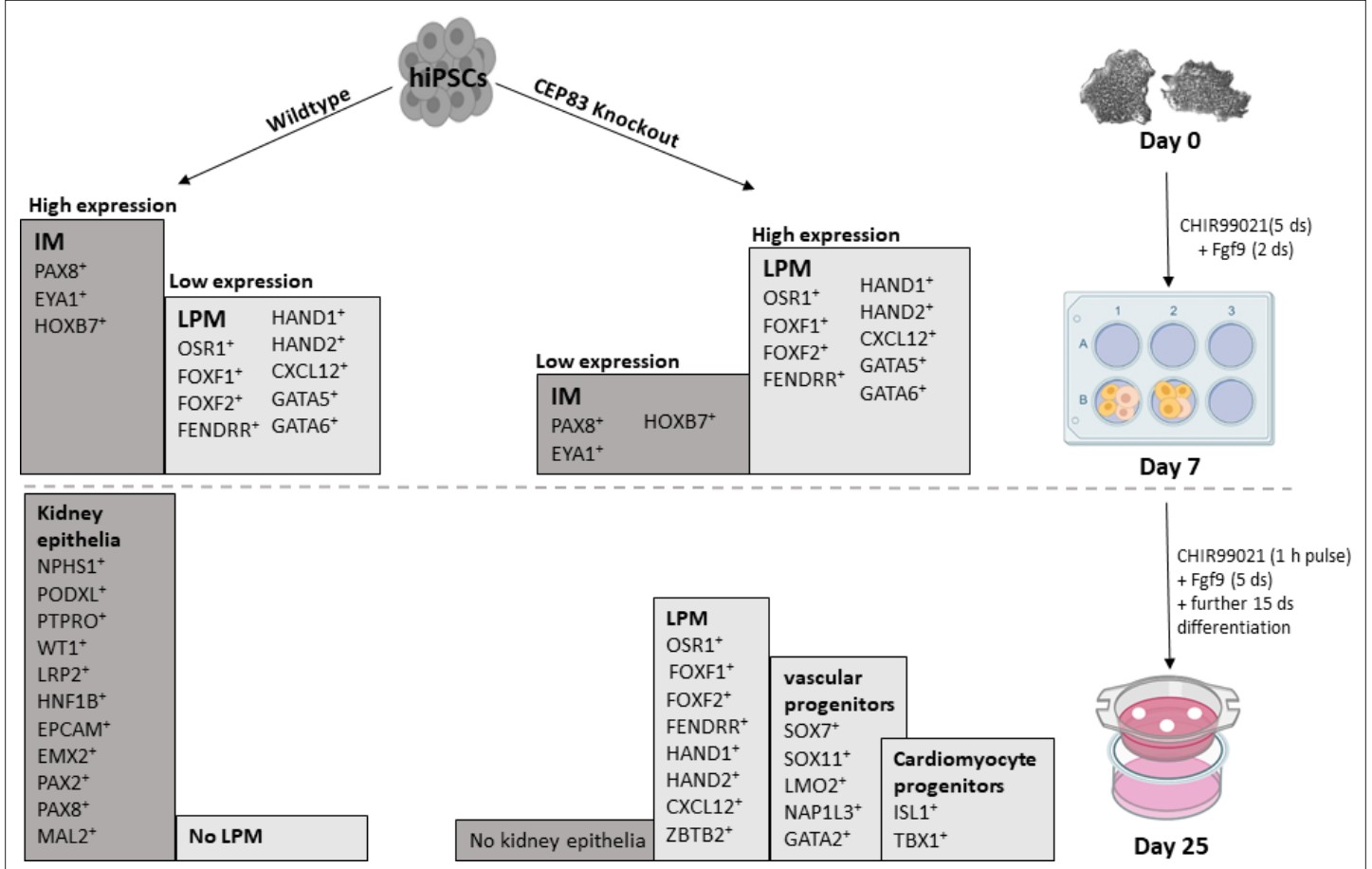

**Figure 8.** Schematic model outlining the functional differences between wildtype and CEP83 knockout cells during the course of differentiation of human pluripotent stem cells toward kidney cells. IM, intermediate mesoderm; LPM, lateral plate mesoderm.

renal agenesis and renal dysplasia (*Filges et al., 2014*; *Reilly et al., 2019*; *Pejskova et al., 2020*). Furthermore, Kif3a, a ciliary protein involved in intraflagellar transport, is necessary for normal mesoderm formation, and kidney progenitor-specific defects of Kif3a have been associated with reduced nephron numbers (*Takeda et al., 1999*; *Chi et al., 2013*). Similarly, mouse genes encoding the ciliary intraflagellar transport proteins IFT25 and IFT27 have been associated with renal agenesis or renal hypoplasia (*Desai et al., 2018*; *Quélin et al., 2018*). Together, these studies highlight the importance of molecules involved in ciliogenesis for mesoderm and kidney progenitor development and suggest that CEP83 contributes to such processes by facilitating an early step of ciliogenesis. Nevertheless, the detailed molecular processes that link CEP83 function, cilia formation, and kidney progenitor specification remain to be determined.

The finding of various upregulated LPM markers in *CEP83^{-/-}* cells starting from day 7 suggests that CEP83 function may be involved in fine-tuning the balance of LPM and IM, thereby contributing to lineage decisions during mesoderm formation. Crosstalk of LPM and IM has been reported previously in zebrafish, overexpression of LPM transcription factors Scl/Tal1 and Lmo2 induces ectopic vessel and blood specification while inhibiting IM formation (*Gering et al., 2003*). Furthermore, the LPM transcription factor Hand2 is critical in determining the size of the IM, while natively expressed in the IM-adjacent LPM progenitors that form mesothelia (*Barbosa et al., 2007Perens et al., 2016*; *Prummel et al., 2022*). Loss of Hand2 in zebrafish results in an expanded IM, whereas Hand2 overexpression reduces or abolishes the IM. Interestingly, HAND2 was among the most strongly induced transcripts in our *CEP83^{-/-}* cells at day 7; connecting with the developmental role of Hand2 in IM formation, these observations suggest that HAND2 expression in CEP83-deficient cells may have contributed to the reduced numbers of nephron progenitor cells at this stage. Of note, CEP83-deficient cells at D25 expressed increased levels of LPM genes expressed in mesothelial (including

*OSR1*, *CXCL12*, *HAND1/2*), cardiopharyngeal (including *ISL1*, *TBX1*), and endothelial/hematopoietic (including *TAL1*, *LMO2*, *GATA2*) progenitors (*Prummel et al., 2020*; *Prummel et al., 2022*). In sum, we propose a novel role for CEP83 in regulating the development of IM nephron progenitors, which may involve direct effects of CEP83 in the nephron progenitor differentiation program and indirect LPM-mediated effects on the IM. Future studies are warranted to delineate the molecular and cellular mechanisms underlying CEP83 function in LPM and specifically IM patterning.

## Materials and methods

### hiPSCs cell line

We used the human iPSC cell line BIHi005-A, which was generated by the Berlin Institute of Health (BIH). The hiPSCs were maintained in six-well plates (Corning, 353046) coated with Matrigel (Corning, 354277) and cultured in Essential 8 medium (E8, A1517001, Gibco-Thermo Fisher Scientific) supplemented with 10 μM Y-27632 (Rocki, Wako, 253–00513). Cells were authenticated and tested for the mycoplasma infection.

### CRISPR CAS9 technology to generate CEP83⁻/⁻ hiPSCs clones

Clustered regularly interspaced short palindromic repeats (CRISPR)-Cas9 technology was used to generate *CEP83⁻/⁻* hiPSCs clones. We designed two CRISPR RNAs (crRNAs) (5'-GGCTGAAG TAGCGGAATTAA-AGG-3' and 5'-AAGAATACAGGTGCGGCAGT-TGG-3') using CRISPOR software (*Concordet and Haeussler, 2018*). The two crRNAs were annealed with trans-activating CRISPR RNA (tracrRNA) to form two guide RNAs (gRNA1 and gRNA2) and then formed a ribonucleoprotein (RNP) complex by incubating gRNA1 and gRNA2 separately with Alt-R S.p. Cas9 Nuclease V3 (1 μM concentration, IDT, 1081058). The hiPSCs were transfected with RNP complexes using Neon transfection system (Thermo Fisher Scientific, MPK5000; *Yumlu et al., 2017*) and Neon transfection 10 μl kit (Thermo Fisher Scientific, MPK10025) according to the manufacturer's instructions. After 48 hr of transfection, we analyzed the editing efficiency in the pool by PCR genotyping.

For PCR genotyping, we isolated genomic DNA from the pool of transfected cells followed by PCR using Phire Tissue Direct PCR Master Mix (Thermo Scientific, F170S) according to the manufacturer's instructions (*Figure 1B*). After confirming the editing efficiency in the pool, we generated single-cell clones by the clonal dilution method. We plated 500 single cells per well of a 6 well plate and picked 24 clones using a picking hood S1 (Max Delbrück Centre Stem Cell Core Facility). Then, clones were screened for homozygous deletions of *CEP83* by PCR using Phire Tissue Direct PCR Master Mix. Selected knockout clones were further characterized for *CEP83* loss of function on the DNA, RNA, and protein level. *CEP83⁻/⁻* clones (*KO1*, *KO2*, and *KO3*) were registered as (BIHi005-A-71, BIHi005-A-72, and BIHi005-A-73) in the European Human Pluripotent Stem Cell Registry (https://hpscreg.eu).

### Single nucleotide polymorphism - karyotype

To assess karyotype integrity, copy number variation (CNV) analysis on the human Illumina OMNI-EXPRESS-8v1.6 BeadChip was used. In brief, genomic DNA was isolated from three *WT* (*WT1*, *WT2*, and *WT3*) and three *KO* (*KO1*, *KO2*, and *KO3*) clones using the DNeasy blood and tissue kit (Qiagen, Valencia, CA, United States), hybridized to the human Illumina OMNI-EXPRESS-8v1.6 BeadChip (Illumina), stained, and scanned using the Illumina iScan system according to a standard protocol (*LaFramboise, 2009*; *Arsham et al., 2017*; *Haraksingh et al., 2017*). The genotyping was initially investigated using the GenomeStudio 1 genotyping module (Illumina). Following that, KaryoStudio 1.3 (Illumina) was used to perform automatic normalization and identify genomic aberrations in detected regions by generating B-allele frequency and smoothed Log R ratio plots. To detect CNVs, the stringency parameters were set to 75 kb (loss), 100 kb (gain), and CN-LOH (loss of heterozygosity). KaryoStudio generates reports and displays chromosome, length, list of cytobands, and genes in CNV-affected regions.

### Differentiation protocol

We used the protocol of *Takasato* to differentiate the hiPSCs into nephron organoids (*Takasato et al., 2015*). Briefly, hiPSCs were cultured first in APEL2 medium (Stem Cell Technologies, 05270) supplemented with 5% Protein Free Hybridoma Medium II (PFHMII, GIBCO, 12040077), and 8 μM CHIR99021 (R&D, 4423/10) for 5 days, with medium changes every 2 days. Then, the cells were cultured in APEL2

medium supplemented with 200 ng/ml FGF9 (R&D, 273-F9-025) and 1 µg/ml heparin (Sigma Aldrich, H4784-250MG) for 2 days. On day 7, the cells were washed with 1× Dulbecco's PBS (DPBS, Thermo Fisher Scientific,14190–250), then trypsinized using trypsin EDTA-0.05% (Thermo Fisher Scientific, 25300–062) at 37°C for 3 min. The cells were counted and divided to achieve $1×10^6$ cells per organoid and cultured into 3D organoid culture on 0.4-µm-pore polyester membrane of Corning 6-well Transwell cell culture plate (Corning-Sigma Aldrich, CLS3450-24EA). Four to five organoids were seeded on one membrane using a P100 wide-bore tip and cultured in APEL2 with 5 µM CHIR99021 at 37°C for 1 hr (CHIR99021 pulse). After the CHIR pulse, we changed the medium to APEL2 medium supplemented with 200 ng/ml FGF9 + 1 µg/ml heparin for 5 days with medium refreshing every 2 days. The organoids were then cultured only in APEL2 medium with 1 µg/ml heparin for additional 13 days. The total differentiation time is 25 days (7+18).

## DNA isolation and PCR

DNA was isolated from cells using DNeasy Blood & Tissue Kits (Qiagen, 69504). CEP83 primers were designed using Primer3 webtool (*Supplementary file 1*). PCR was done using Phusion high-fidelity DNA polymerase (Biolabs, New England, M0530) according to the manufacturer's instructions. PCR results were visualized on 1.5% agarose gel using a BioDoc Analyze dark hood and software system (Biometra).

## RNA isolation, RNA sequencing, and qPCR

Total RNA was isolated from the cells using RNAasy Mini Kit (QIAGEN, Hilden, Germany, 74104) following the manufacture's instructions. The concentration, quality, and integrity of the extracted RNA were evaluated using Nanodrop (Thermo Scientific, Waltham, MA; USA), an Agilent 2100 Bioanalyzer, and the Agilent RNA 6000 Nano kit (Agilent Technologies, 5067–1511). 0.4 µg total RNA was used to obtain a poly A–enriched RNA library by Novogene (Cambridge, United Kingdom). Library concentration was performed using a Qubit fluorometer (HS RNA assay kit, Agilent Technologies). Library size was measured by Agilent 2100 bioanalyzer. The libraries were then subjected to 150 bp paired-end next-generation sequencing (Illumina NovaSeq 6000 S4 flow cells). Mutation visualization was performed using the Integrative Genomic Viewer tool (*Robinson et al., 2011*). Read counts of the sequenced RNA were normalized to transcripts per million (TPM). The TPM values of the variables were used to plot heatmaps and for principle component analysis (PCA) based on Pearson correlation, using self-written scripts in R (R Development Core Team (2011)) (version 4.0.4).

RNA was reverse transcribed using the RevertAid First Strand cDNA Synthesis Kit (Thermo Scientific). qPCR was performed using the FastStart Universal SYBR Green Master (Rox) mix (Hoffmann-La Roche) according to the manufacturer's instructions. Glyceraldehyde-3-phosphate dehydrogenase (*GAPDH*) mRNA expression was calculated according to the ΔΔCt method. All primer pairs were designed using Primer3, purchased at BioTeZ (Berlin, Germany), and sequences are shown in *Supplementary file 1*.

## Single-cell RNA sequencing (scRNA-seq)

### Cells isolation and preparation

Differentiated cells at day 7 were washed twice with 1× DPBS, dissociated with Accumax solution, and resuspended in 1× DPBS. Then, cells were filtered, counted, and checked for viability.

### Library preparation and single-cell sequencing

Single-cell 3' RNA sequencing was performed using the 10× Genomics toolkit version v3.1 (*Alles et al., 2017*), according to the manufacturer's instructions aiming for 10,000 cells. Obtained libraries were sequenced on Illumina NextSeq 500 sequencers.

### Single-cell sequencing data analysis and clustering

After sequencing and demultiplexing, fastq files were analyzed using Cellranger version 3.0.2. Gene expression matrices were then imported into R, and Seurat objects were created using the Seurat R package (version 4.0.5) (*Stuart et al., 2019*). The gene expression matrices were initially filtered by applying lower and upper cut-offs for the number of detected genes (500 and 6000, respectively). The filtered data were then log normalized and scaled according to the number of unique molecular

identifiers. The normalized and scaled data derived from the four samples were then merged into one Seurat object. Clustering was performed using the first 20 principal components. We used the Seurat FindAllMarkers function to extract marker gene lists that differentiate between clusters with log fold change threshold ± 0.25 using only positive markers expressed in a minimum of 25% of cells. PCA was done using the first 20 principle components in R using the following libraries factoextra, FactoMineR, and ggplot2.

## Protein extraction and immunoblotting

Proteins were extracted from hiPSCs using radioimmunoprecipitation assay (RIPA) buffer (Sigma-Aldrich, R0278) as described in details in supplementary data. 30 µg protein in RIPA buffer were mixed with 1× reducing (10% b-mercaptoethanol) NuPAGE loading buffer (Life Technologies, Carlsbad, CA), loaded on a precast polyacrylamide NuPage 4–12% Bis-Tris protein gel (Invitrogen, Carlsbad, CA, USA) and blotted on 0.45 µm pore size Immobilon-P Polyvinylidene difluoride membrane (EMD Milli-pore, Billerica, MA; USA). The membrane was blocked in 5% bovine serum albumin for 1 hr at room temperature and incubated overnight at 4°C with primary antibodies: anti-CEP83 produced in rabbit (1:500, Sigma-Aldrich) and anti-α-Tubulin produced in mouse (1:500, Sigma-Aldrich, T9026). Then, the membrane was incubated for 1 hr at room temperature with horseradish peroxidase-conjugated secondary antibodies (1:2000, Sigma-Aldrich, Saint Louis, MO, USA). Chemiluminescent reagent (Super Signal–West Pico; Thermo Scientific, Waltham, MA; USA) was used to detect the proteins. The spectra Multicolor Broad Range Protein Ladder (Thermo Fisher Scientific, USA) was used to evaluate the molecular weight of corresponding protein bands.

## Histology and immunofluorescence staining

Cells at different time points were checked regularly under a confocal microscope (Leica DMI 6000 CEL) for differentiation progress. Quantitative analysis of nephron-like structure formation within each organoid (D25) was done on tile scanning images of each organoid by estimating the percentage of the organoid area composed of nephron-like structures using 13 WT and 9 KO organoids. Organoids were fixed in BD Cytofix buffer (554655, BD Biosciences) for 1 hr on ice. Then organoids were grad-ually dehydrated in increasing ethanol concentrations, cleared in xylene, and embedded in paraffin. Organoids were cut into 3.5-µm thick sections. The sections were deparaffinized, dehydrated, and stained in hematoxylin (Sigma-Aldrich, Saint Louis, MO) for 3 min and in 1% eosin (Sigma-Aldrich) for 2 min. For immunostaining, cells (day 7) and organoids (day 25) were fixed with BD Cytofix, permea-bilized with BD Perm/Wash (554723, BD Biosciences), and blocked with blocking solution (1% BSA + 0.3% triton-X-100 in 1× DPBS) for 2 hr. Cells and organoids were incubated overnight at 4°C with primary antibodies (*Supplementary file 2*), then incubated with fluorescence-labeled secondary anti-bodies with 1:500 dilution including Cy3, Cy5, Alexa488, and Alexa647 (Jackson ImmunoResearch, Newmarket, UK) and Cy3 Streptavidin (Vector lab, Burlingame, USA) overnight at 4°C. DAPI was then used for nuclear staining (Cell signaling Technology, Danvers, MA, USA) with 1:300,000 dilution for 1 hr at RT. Finally, cells were mounted with Dako fluorescent mounting medium (Agilent Technolo-gies). Images were taken using a SP8 confocal microscope (Carl Zeiss GmbH, Oberkochen, Germany). Quantitative analyses of acquired images were performed using ImageJ software (1.48 v; National Institutes of Health, Bethesda, MD).

## Comparison to zebrafish LPM

The upregulated genes in CEP83−/− cells on day 7 and day 25 were compared with the top 20 orthol-ogous genes identified in subclusters of zebrafish LPM identified by scRNA-seq, as deposited on ArrayExpress (E-MTAB-9727; *Prummel et al., 2022*).

## Statistical analysis

scRNA-seq was done on two biological replicates representing two different clones of CEP83−/− and control cells, respectively. All other experiments were performed using three biological replicates representing three independent clones of *CEP83*−/− and control cells at different time points. A common excel sheet for the genes present in both bulk RNA and scRNA sequencing was generated in R. The sheet includes a total of 20,894 genes and represents the TPM values of both groups (*WT* and *KO*) on day 0, day 7, and day 25 of differentiation. The maximum TPM (TPMmax) and the minimum

TPM (TPMmin) were calculated for each gene across all samples. HVGs were calculated based on the ratio of TPMmax and TPMmin. For heatmaps and PCA analysis, the top 1000 HVGs were plotted with a selection of TPMmax >2 for each gene. Deregulated (upregulated and downregulated) genes between *WT* and *KO* groups were selected with expression criteria of TPM >2, fold change >1.5, and p-value calculated on log10 TPM <0.05. The unpaired two-tailed t-test was used to compare the two groups. All graphs were generated using GraphPad Prism 7.04 (GraphPad Software, San Diego, CA). Data are presented as mean ± SD.

## Acknowledgements

We thank Tatjana Luganskaja for her excellent technical support. This work was supported by grants to KMS-O from the Deutsche Forschungsgemeinschaft (DFG; SFB 1365, GRK 2318, and FOR 2841), by stipends to FM by the Egyptian government, by the Urological Research Foundation (Berlin), a Swiss National Science Foundation postdoctoral fellowship to JK-R, and the University of Colorado School of Medicine, Anschutz Medical Campus, and the Children's Hospital Colorado Foundation to CM.

## Additional information

### Funding

| Funder | Grant reference number | Author |
| --- | --- | --- |
| Deutsche Forschungsgemeinschaft | GRK 2318 | Kai M Schmidt-Ott |
| Deutsche Forschungsgemeinschaft | FOR 2841 | Kai M Schmidt-Ott |
| Deutsche Forschungsgemeinschaft | SFB 1365 | Kai M Schmidt-Ott |
| National Science Foundation | | Jelena Kresoja |
| University of Colorado | | Christian Mosimann |

The funders had no role in study design, data collection and interpretation, or the decision to submit the work for publication.

### Author contributions

Fatma Mansour, Software, Formal analysis, Validation, Visualization, Methodology, Writing - original draft; Christian Hinze, Narasimha Swamy Telugu, Jelena Kresoja, Methodology; Iman B Shaheed, Writing – review and editing; Christian Mosimann, Methodology, Writing – review and editing; Sebastian Diecke, Supervision, Methodology; Kai M Schmidt-Ott, Conceptualization, Supervision, Writing – review and editing, Funding acquisition

### Author ORCIDs

Fatma Mansour http://orcid.org/0000-0003-4808-3514
Christian Hinze http://orcid.org/0000-0003-2526-1621
Christian Mosimann http://orcid.org/0000-0002-0749-2576
Kai M Schmidt-Ott http://orcid.org/0000-0002-7700-7142

### Decision letter and Author response

Decision letter https://doi.org/10.7554/eLife.80165.sa1
Author response https://doi.org/10.7554/eLife.80165.sa2

## Additional files

### Supplementary files
• MDAR checklist

- Supplementary file 1. The table shows the primers list used in the qPCR.
- Supplementary file 2. The table shows the list of primary antibodies used in IF staining.

## Data availability

All data supporting the findings of this study are available within the article and its supplementary files. Source data files have been provided for Figures 1–6. Sequencing data have been deposited in GEO under accession code GSE205978.

The following dataset was generated:

| Author(s) | Year | Dataset title | Dataset URL | Database and Identifier |
|---|---|---|---|---|
| Schmidt-Ott KM | 2022 | The centrosomal protein 83 (CEP83) regulates human pluripotent stem cell differentiation toward the kidney lineage | http://www.ncbi.nlm.nih.gov/geo/query/acc.cgi?acc=GSE205978 | NCBI Gene Expression Omnibus, GSE205978 |

The following previously published dataset was used:

| Author(s) | Year | Dataset title | Dataset URL | Database and Identifier |
|---|---|---|---|---|
| Hariharan K | 2019 | Parallel generation of easily selectable multiple nephronal cell types from human pluripotent stem cells | https://www.ncbi.nlm.nih.gov/geo/query/acc.cgi?acc=GSE75711 | NCBI Gene Expression Omnibus, GSE75711 |

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

## Appendix 1

### Supplemental methods

#### hiPSCs culture

We used the human iPSC cell line BIHi005-A, which was generated from a healthy donor by the Berlin Institute of Health (BIH) and supplied by the stem cell core facility at Max Delbrück Center for Molecular Medicine (Berlin). The hiPSCs were maintained in six-well plates (Corning, 353046), coated with Matrigel (Corning, 354277), and cultured in Essential 8 medium (E8, Gibco-Thermo Fisher Scientific, A1517001) supplemented with 10 μM Y-27632 (Rocki, Wako, 253–00513). The cells were split twice per week using EDTA/PBS or Accumax (Stem cell technology, 07921).

#### CRISPR CAS9 technology to generate CEP83$^{-/-}$ hiPSCs clones

Clustered regularly interspaced short palindromic repeats (CRISPR)-Cas9 technology was used to generate CEP83$^{-/-}$ hiPSCs clones. Two 20 bp-long CRISPR RNAs (crRNAs) were designed using CRISPOR software (*Concordet and Haeussler, 2018*) to selectively target exon7: (5'- GGCTGAAG TAGCGGAATTAA-AGG-3'); (5'-AAGAATACAGGTGCGGCAGT-TGG-3'). The crRNAs were ordered from Integrated DNA Technologies (IDT). The crRNAs (IDT) were annealed in equimolar concentrations with trans-activating CRISPR RNA (tracrRNA) to form two guide RNAs (gRNA1 and gRNA2), which were then conjugated separately with Alt-R S.p. Cas9 Nuclease V3 (1 μM concentration, IDT, 1081058) at room temperature for 1 hr to form ribonucleoprotein (RNP) complexes (RNP1 and RNP2).

One day prior to transfection, hiPSCs were split using Accumax solution and cultured in an equal proportion of E8 medium and StemFlex medium (Thermo Fisher Scientific, A3349401). The hiPSCs were transfected using a Neon transfection system (Thermo Fisher Scientific, MPK5000). Immediately before the transfection, the cells were dissociated, collected, and resuspended in Resuspension Buffer (Buffer R) that included in Neon transfection 100 μl kit (Thermo Fisher Scientific, MPK10025; *Yumlu et al., 2017*). Cells were transfected in 3 ml Electrolytic buffer (Buffer E) that included in the neon transfection kit and by using the Neon transfection system 10 μl tip. The used Neon Transfection parameters were voltage (1200 V), width (30 ms), and pulse (1). The transfected cells were cultured in StemFlex Medium with Rocki for 48 hr.

Then, quick DNA extraction and PCR were then done to test transfection efficiency according to the manufacturer's instructions using Phire Tissue Direct PCR Master Mix (Thermo Scientific, F170S). The size of the PCR products was visualized on 1.5% agarose gel. After confirming the transfection's success in the knockout cells, as shown in *Figure 1B*, the cells were dissociated and seeded at low densities for 24 hr. Then, 24 single cells were picked under a picking hood S1 (stem cell core facility-MDC, Buch) and cultured in StemFlex medium for 2 weeks. The clones were then tested for CEP83 mutation on the DNA level by PCR using Phire Tissue Direct PCR Master Mix. Finally, the selected clones were expanded and frozen in the Bambanker medium (Nippon Genetics, BB01) for further characterization. The selected clones were characterized for the mutation induction on the DNA, protein, and RNA level.

#### Differentiation protocol

We used the protocol of *Takasato* to differentiate the hiPSCs into nephron organoids (*Takasato et al., 2015*), the experiment was performed using three replicates per each group wildtype hiPSCs (*WT1*, *WT2*, and *WT3*) and CEP83$^{-/-}$ hiPSCs (*KO1*, *KO2*, and *KO3*). Two days prior to the differentiation, cultured hiPSCs on matrigel with 70–80% density were prepared for the differentiation. The cells were washed twice with 1× Dulbecco's PBS (Thermo Fisher Scientific,14190–250), then cells were trypsinized using 1× Trypl E Select (Thermo Fisher Scientific, 12563011). Cells were incubated at 37°C for 3 min. Then, DMEM/F-12 medium (Thermo Fisher Scientific, 11320–033) was added to the cells to neutralize Trypl E. The cell suspension was mixed by pipetting (pipetting is maximum twice), then centrifuged at 300 g for 5 min. The cell pellet was washed and resuspended in 1 ml of E8 medium. Then the cells were counted using Countess chamber slide and Countess II Automated Cell Counter (Thermo Fisher Scientific). Then cells were centrifuged and resuspended in E8 media supplemented with 10 μM Rocki. Last, cells were cultured on a prepared coated matrigel six-well culture plates to obtain a density of $15×10^3$ cells per cm$^2$ and incubated overnight at 37°C CO$_2$ incubator for 48 hr with daily medium change.

Immediately before the differentiation, the cells were checked under the microscope. Cells with 40–50% confluency were used for the differentiation. The E8 medium was changed into APEL2

medium (Stem Cell Technologies, 05270) with 5% Protein Free Hybridoma Medium II (PFHMII, GIBCO, 12040077) and 8 µM CHIR99021 (2 ml medium per a well of 6- well plate). Cells were incubated in a 37°C CO$_2$ incubator for 5 days, with medium refreshing every 2 dys. Following the CHIR99021 phase, the medium was changed into a double volume of APEL2 medium (4 ml medium per a well of six-well plate) supplemented with 200 ng/ml FGF9 (R&D, 273-F9-025) and 1 µg/ml heparin (Sigma Aldrich, H4784-250MG) and were incubated in a 37°C CO$_2$ incubator.

On day 7 of differentiation, the cells were washed, trypsinized with trypsin EDTA (0.05%), and incubated at 37°C for 3 min. Then, the cell suspension was transferred to a 50 ml tube containing 9 ml of MEF conditioned medium (R&D, AR005) to neutralize the trypsin. The cells were centrifuged and resuspended in APEL2 medium. Using a hemocytometer, the cells were counted, and the cell suspension was divided to achieve $1 \times 10^6$ cells (organoid) per 1.5 ml tube. All the tubes were centrifuged at 400× g for 3 min at RT. During centrifugation, six-well transwell cell culture plates (Corning-Sigma Aldrich, CLS3450-24EA) were prepared by adding 1.2 ml of APEL2 supplemented with 5 µM CHIR99021 to each well. Cell pellets were picked up by using a P1000 or P200 wide-bore tip. Pellets were carefully seeded onto the six-well transwell membrane with minimal APEL2 medium carryover and incubated at 37°C for 1 hr. Then the medium was changed into APEL2 medium supplemented with 200 ng/ml FGF9 plus 1 µg/ml heparin for further 5 days, with medium refreshing every 2 days. Finally, the medium to APEL2 medium with only heparin for further 13 days, with medium refreshing every 2 days.

## DNA isolation and PCR

Cultured *WT* and *CEP83*$^{-/-}$ hiPSCs were washed, scrapped gently using a cell scraper (VWR, part of Avantor, 734–2602), and collected with a maximum $5 \times 10^6$ cell/ml for proper DNA extraction. The DNA was extracted using DNeasy Blood & Tissue Kits (Qiagen, 69504) following the manufacturer's instructions. The concentrations and quality of the DNA were evaluated using Nanodrop (Thermo Scientific, Waltham, MA; USA). To detect CEP83 expression, 200 µg DNA was amplified by a standard PCR using Phusion high-fidelity DNA polymerase (Biolabs, New England, M0530). The master mix was calculated according to the manufacturer's instructions. Primers are designed using Primer3 webtool, ***Supplementary file 2***. PCR was carried out in a thermocycler as follow: initial denaturation at 98°C for 30 s, 35–40 cycles of 30 s at 98°C, 30 s at 63.5°C, and 30 s at 72°C; final elongation step at 72°C for 10 min. The PCR results were checked on 1.5% agarose gel and analyzed using a BioDoc Analyze dark hood and software system (Biometra).

## RNA isolation, RNA sequencing, and qPCR

Total RNA was isolated from the cells at three time points: day 0 (hiPSCs), day 7 (IM), and day 25 (organoids) with a maximum of $1 \times 10^7$ cells using RNAasy Mini Kit (QIAGEN, Hilden, Germany, 74104,) following the manufacture instructions. The RNA was treated with RNase-free DNase I (QIAGEN, 79254) for 15 min at room temperature during the extraction. The concentration, quality, and integrity of the extracted RNA were evaluated using Nanodrop (Thermo Scientific, Waltham, MA; USA), an Agilent 2100 Bioanalyzer, and the Agilent RNA 6000 Nano kit (5067–1511, Agilent Technologies). More than 0.4 µg total RNA with high integrity (more than 6.8) and high purity (OD260/280=1.8–2.2 and OD260/230≥1.8) were collected and sent to Illumina NovaSeq 6000RNA sequencing by Novogene. RNA-Seq library preparation and next-generation sequencing: cDNA libraries with paired-end 150 bp enriched were prepared by Novogene. First, the mRNA was randomly fragmented and supplemented with oligo (dT) beads. Then cDNA synthesis was done using the random hexamers and reverse transcriptase. Second, second-strand synthesis was done using: a custom second-strand synthesis buffer from Illumina, deoxyribose nucleoside triphosphates (dNTPs), RNase H, and *Escherichia coli* polymerase I. The final obtained cDNA library was purified, terminally repaired, A-tailed ligated to sequencing adapters, size-selected, and PCR-enriched. Quantification of library concentration was performed using a Qubit 2.0 fluorometer. Library size was measured by Agilent 2100 bioanalyzer and was quantified by qPCR (library activity >2 nM). Libraries were sequenced on Illumina NovaSeq 6000 S4 flow cells (paired end, 150 bp).

Raw data were transformed into sequenced reads and recorded in a FASTQ file. FASTQ files were aligned to build 19 of the human genome provided by the Genome Reference Consortium (GRCh19), performed by Christian Hinze using TOPHAT2 aligner tool (***Kim et al., 2013***). Up to four mismatches with the reference genome were accepted. Raw counts were obtained using featureCounts (***Liao et al., 2014***). Mutation visualization in the knockout samples was performed using the Integrative

Genomic Viewer tool (**Robinson et al., 2011**). For gene expression analysis reads were normalized to the sequence length and transcripts per million (TPM) values were calculated (**Wagner et al., 2012**). TPM values of the samples were used to plot heatmaps and for PCA analysis based on Pearson correlation, using R (R Development Core Team, 4.0.4)

500 ng of RNA was reverse transcribed using the RevertAid First Strand cDNA synthesis kit (Thermo Scientific, K1622) according to the manufacturer's instructions. The qPCR was carried out using the Fast Universal SYBR Green Master Mix (ROX, Roche Diagnostics, 04 913 850 001,) according to the manufacturer's instructions. For expression analysis, relative mRNA expression levels were normalized for GAPDH mRNA expression and calculated according to the ΔΔCt method. All primer pairs were designed using the free-online primer design tool Primer3, purchased at BioTeZ (Berlin, Germany), and sequences are shown in (**Supplementary file 1**). The statistical significance of differences between two groups (WT and KO) was analyzed using a two-sided Student's t-test.

## Single-cell RNA (scRNA) experiment
### Cells isolation and preparation
The differentiated hiPSCs to intermediate mesodermal cells were collected at day 7 of the differentiation from two different experiments. The cells of the first experiment were derived from *WT1* and *KO1* differentiated hiPSCs, while the second experiment comprises the differentiated cells of *WT2* and *KO2* cells. The cells were washed twice with 1× DPBS and dissociated with Accumax for 7 min at 37 °C. Cells were centrifuged at 350× g for 5 min and resuspended in 1× DPBS. Then, cells were filtered with a 40 µm filter (Corning, 352340), counted (10,000 cells per sample), and checked for viability using Trypan blue staining.

## Protein extraction and Immunoblotting
### Protein extraction

Up to 1×10$^6$ hiPSCs per sample (*WT1*, *WT2*, *WT3*, *KO1*, *KO2*, and *KO3*) were washed with cold 1× PBS, then centrifuged at 3500 g for 5 min. Next, the cell pellet was resuspended in pre-ice cold 100 µl of radioimmunoprecipitation assay (RIPA) buffer (Sigma-Aldrich, R0278) supplemented with protease inhibitor (Roche, 11697498001) and maintained with constant agitation for 30 min at 4°C. Then the suspension was centrifuged at 4°C for 20 min at 12,000 rpm. The supernatant was collected as protein extract and quantified using BCA Protein Assay (Thermo Scientific, 23228).

### Immunoblotting

30 µg of the extracted protein in RIPA buffer were mixed with 1RIPA reducing (10% b-mercaptoethanol) NuPAGE loading buffer (Life Technologies, Carlsbad, CA). After denaturation at 70°C for 10 min. The protein was loaded on a precast polyacrylamide NuPage 4–12% Bis-Tris protein gel (Invitrogen, Carlsbad, CA, USA) and 1× MOPS (1 M MOPS, 1 M TrisBase, 69.3 mM SDS, 20.5 mM EDTA) to be separated according to the length using SDS -PAGE (100 V, 200mA, 2 hr). Proteins were blotted on 0.45 µm pore size Immobilon-P polyvinylidene difluoride membrane (EMD Millipore, Billerica, MA; USA). The membrane was pre-activated for 20 s in methanol and equilibrated in 1× NuPage Transfer buffer (1.25 mM bicine, 1.25 mM BisTris, 0.05 mM EDTA, and 10% ethanol) for 30 min at RT. The membrane was blocked in 5% bovine serum albumin for 1 hr at RT and incubated overnight at 4°C with primary antibodies: Anti-CEP83 produced in rabbit (1:500, Sigma-Aldrich) and Anti-α-Tubulin produced in mouse (1:500, T9026, Sigma-Aldrich). The membrane was incubated for 1 hr at RT with horseradish peroxidase-conjugated secondary antibodies (Sigma-Aldrich, Saint Louis, MO, USA) with 1:2000 dilution. Chemiluminescent reagent (Super Signal–West Pico; Thermo Scientific, Waltham, MA; USA) was used to detect the proteins. The spectra Multicolor Broad Range Protein Ladder (Thermo Fisher Scientific, USA) was used to evaluate the molecular weight of corresponding protein bands.

## Histology and immunofluorescence staining
After organoid fixation in BD Cytofix buffer (554655, BD Biosciences) for 1 hr on ice, the organoids were gradually dehydrated in increasing ethanol concentrations for 15 min each. Then organoids were cleared in xylene for three times 20 min each. After infiltration with melted paraffin at 65°C

three times for 30 min each, the organoids were embedded in paraffin and processed in 3.5-µm thick sections using a HM 355 S microtome. The sections were deparaffinized, dehydrated, and stained with hematoxylin (Sigma-Aldrich, Saint Louis, MO) for 3 min and in 1% eosin (Sigma-Aldrich) for 2 min. The sections were mounted using Kaiser's glycerol gelatin-based mounting medium. Images were captured with a Leica CTR 6000 microscope (Leica Biosystems, Wetzlar, Germany).

For immunostaining, cultured cells (D7) and organoids (D25) were fixed with BD Cytofix for 10 min on ice. Then cells were permeabilized with BD Perm/Wash (554723, BD Biosciences), twice, 15 min per each. Cells were blocked with a blocking solution (1% BSA + 0.3% triton-X-100 in 1× DPBS) for 2 hr at RT or overnight at 4°C. Cells were incubated overnight at 4°C with primary antibodies (*Supplementary file 2*). Cells were then washed twice (10 min each) and incubated with fluorescence-labeled secondary antibodies with 1:500 dilution including Cy3, Cy5, Alexa488, and Alexa647 (Jackson ImmunoResearch, Newmarket, UK) and Cy3 Streptavidin (Vector lab, Burlingame, USA) overnight at 4°C. DAPI was then used for nuclear staining (Cell signaling Technology, Danvers, MA, USA) with 1:300,000 dilution for 1 hour at RT. Finally, cells were mounted with Dako fluorescent mounting medium (Agilent Technologies). The images were taken using a SP8 confocal microscope (Carl Zeiss GmbH, Oberkochen, Germany). All the quantitative analyses of the taken images were performed using ImageJ (1.48 v; National Institutes of Health, Bethesda, MD) software.

