## [Editor Report]

This paper describes a novel role of the centrosomal protein CEP83 in mesoderm patterning. Specifically, CEP83 is shown to be required for the formation of the intermediate mesoderm and kidney progenitor tissue derived from it. In a CEP83 knockout situation in human pluripotent stem cells, a shift to lateral plate mesoderm at the expense of intermediate mesoderm occurs, which is convincingly demonstrated. This work, therefore, presents important new insights into the processes involved in mesodermal lineage induction and the fine-tuning of kidney differentiation.

---

## [Decision Letter]

**Decision letter after peer review:**

Thank you for submitting your article "The centrosomal protein 83 (CEP83) regulates human pluripotent stem cell differentiation towards the kidney lineage" for consideration by *eLife*. Your article has been reviewed by 2 peer reviewers, including Veronika Sander as the Reviewing Editor and Reviewer #1, and the evaluation has been overseen by Didier Stainier as the Senior Editor.

Essential revisions:

1) Add discussion and data (if possible) around the questions raised by Reviewer #2, e.g. trajectories towards LPM fate, statistics to the clusters, and thoughts on whether the phenotype is related to the ciliary function of CEP83.

2) Address several formal issues regarding the presentation of some of the results and the description of the methods.

*Reviewer #1 (Recommendations for the authors):*

Listed here are my concerns and comments. If addressed adequately, I recommend this work for publication in *eLife*.

1. The description of the methods should be restricted to the 'Methods' section. Right now, there is a lot of technical detail in the figure legends for e.g. Figure 1 and 4. Please revise and shorten the Methods and keep them strictly separate from the other sections of the manuscript.

2. The numbering of the supplementary figures does not match the main text. This makes it hard to keep track when reading the manuscript.

3. Some of the supplementary data is described and interpreted exhaustively, e.g. Figure 5—figure supplement 1 (the dot-plot figure). Most of the information in this figure's legend is not relevant. I suggest reducing the figure to the clusters of interest, i.e. 0, 1, 3, 4, and shortening the legend.

4. I wonder if the quantitative measurements in Figure 3, 5, and 6 could be better presented as bar graphs. I find that the dot and violin plots don't show the differences as obviously as bar graphs would do, and no additional information is gained from e.g. the violin plots in Figure 5.

5. The organoid data presented in Figure 3 are impressive but the significance of this set of data could be improved by adding more meaningful and convincing immunostainings, e.g. Vimentin or markers of the LPM (as identified by the RNAseqs) that proposedly would label the cells in the CEP83 sections should be shown. Panel 3E could be replaced with a less broken section and more intact-looking LTL+ tubules. Also, the official gene/protein name of ECAD is CDH1.

6. Out of curiosity, has the spontaneous beating of CEP83-knockout organoids, indicative of cardiomyocyte formation in these structures, been observed?

7. The brightfield images of the whole organoids in Figure 3 should also be replaced with images that more clearly show the differences, e.g. I cannot recognize 'kidney-like' structures (tubules? Podocyte clusters?) in Figure 3A. Panel 3B seems similar to 3A but out of focus are more necrotic. Please replace these panels, maybe include high magnification panels.

8. Figure 4D could be supplementary, and could you instead show a UMAP plot of the clusters of interest, 0, 1, 3, and 4?

9. Could Figure 6 be split into 2 figures for better clarity?

10. Figure 6 J/O, one panel for ISL1 expression is sufficient.

11. I also recommend a general revision of English style and grammar, as well as a check for consistency, e.g. is intermediate mesoderm abbreviated as IM or IMM? Some typos I noticed:

• P7, line 18, under a confocal microscope

• P7, line 20, was done

• P7, line 26, should this read 'rehydrated'?

• P7, lines 29 and 35, cells? Should this be 'organoids' or 'organoid sections'?

• P8, line 10, was generated

*Reviewer #2 (Recommendations for the authors):*

I enjoyed reading the manuscript and found it insightful and logically structured. The main limitation I see with this manuscript is that it is rather short on mechanistic insights. It is tempting to speculate that the wealth of data generated may contain some hints as to what signalling events may drive the LPM fate? Could trajectories help in this regard or are there technical limitations that prohibit this? And a major question that remains is if the phenotype is due to a ciliary defect or extraciliary function of CEP83.

It is interesting that Osr1 is upregulated in the CEP83-/- cells, but what about other more traditional nephron progenitor makers, such as SIX2 or CITED? The percentage of contributing cells to the clusters is interesting, but can this data be boosted by statistics to evaluate if this is literally significant or not? I noticed that cluster 5 (damaged cells) seemed to be larger in the KO condition. What kind of damage does this entail and to what degree might this influence the broader interpretation of the experiment?

---

## [Author Response]

Essential revisions:1) Add discussion and data (if possible) around the questions raised by Reviewer #2, e.g. trajectories towards LPM fate, statistics to the clusters, and thoughts on whether the phenotype is related to the ciliary function of CEP83.

We added data and discussion as specified in responses to reviewer 2.

2) Address several formal issues regarding the presentation of some of the results and the description of the methods.

This was done. See below.

Reviewer #1 (Recommendations for the authors):Listed here are my concerns and comments. If addressed adequately, I recommend this work for publication in eLife.1. The description of the methods should be restricted to the 'Methods' section. Right now, there is a lot of technical detail in the figure legends for e.g. Figure 1 and 4. Please revise and shorten the Methods and keep them strictly separate from the other sections of the manuscript.

We modified the figure legends as requested.

2. The numbering of the supplementary figures does not match the main text. This makes it hard to keep track when reading the manuscript.

We apologize for any confusion this may have caused. The supplemental figures are now numbered according to journal instructions.

3. Some of the supplementary data is described and interpreted exhaustively, e.g. Figure 5—figure supplement 1 (the dot-plot figure). Most of the information in this figure's legend is not relevant. I suggest reducing the figure to the clusters of interest, i.e. 0, 1, 3, 4, and shortening the legend.

We agree that this text did not add to the key message of the supplemental figure. We shortened the figure legend accordingly. To ensure completeness of the figure, we maintained all clusters, but highlighted the relevant clusters 1, 3, and 4.

4. I wonder if the quantitative measurements in Figure 3, 5, and 6 could be better presented as bar graphs. I find that the dot and violin plots don't show the differences as obviously as bar graphs would do, and no additional information is gained from e.g. the violin plots in Figure 5.

Thank you for this comment. We think that the advantage of the violin plots is that they not only show the differences in gene expression but also the differences in cell numbers expressing these genes. It is true that the difference in e. g. HOXB7 expression is not as obvious in violin plots, but the dot plot in the newly modified Figure 5: Suppl. Figure 1 now clearly shows differential expression of HOXB7 as well. We therefore hope that the reviewer agrees with our decision to maintain violin plots in the main figures.

5. The organoid data presented in Figure 3 are impressive but the significance of this set of data could be improved by adding more meaningful and convincing immunostainings, e.g. Vimentin or markers of the LPM (as identified by the RNAseqs) that proposedly would label the cells in the CEP83 sections should be shown. Panel 3E could be replaced with a less broken section and more intact-looking LTL+ tubules. Also, the official gene/protein name of ECAD is CDH1.

We thank the reviewer for these helpful suggestions. A new supplementary Figure (Figure 3—figure supplement 1) was added to show a better LTL staining in the WT organoids. The label ECAD on the figure was changed to CDH1.

6. Out of curiosity, has the spontaneous beating of CEP83-knockout organoids, indicative of cardiomyocyte formation in these structures, been observed?

Thank you for this interesting question. However, we did not observe spontaneously beating structures.

7. The brightfield images of the whole organoids in Figure 3 should also be replaced with images that more clearly show the differences, e.g. I cannot recognize 'kidney-like' structures (tubules? Podocyte clusters?) in Figure 3A. Panel 3B seems similar to 3A but out of focus are more necrotic. Please replace these panels, maybe include high magnification panels.

Figure 3A and 3B were changed to provide higher magnification images.

8. Figure 4D could be supplementary, and could you instead show a UMAP plot of the clusters of interest, 0, 1, 3, and 4?

Since UMAPs are unbiased, the only way of restricting the analysis to clusters 0,1,3 and 4 would be to select these cells and subcluster them. However, this analysis did not add additional information beyond what is already provided by the unbiased UMAP. We therefore decided to leave the figure as is. The main point of Figure 4D is to show that we observed no substantial differences in cluster distribution across the four samples (with noted exceptions)

9. Could Figure 6 be split into 2 figures for better clarity?

As recommended, we split Figure 6 (New Figure 6 and New Figure 7).

10. Figure 6 J/O, one panel for ISL1 expression is sufficient.

We apologize for the oversight. This was changed accordingly.

11. I also recommend a general revision of English style and grammar, as well as a check for consistency, e.g. is intermediate mesoderm abbreviated as IM or IMM? Some typos I noticed:• P7, line 18, under a confocal microscope• P7, line 20, was done• P7, line 26, should this read 'rehydrated'?• P7, lines 29 and 35, cells? Should this be 'organoids' or 'organoid sections'?• P8, line 10, was generated

Thank you for these comments. All changes were implemented and the manuscript was proof-read again by a native English speaker.

Reviewer #2 (Recommendations for the authors):I enjoyed reading the manuscript and found it insightful and logically structured.

We thank the reviewer for the positive assessment of our paper.

The main limitation I see with this manuscript is that it is rather short on mechanistic insights. It is tempting to speculate that the wealth of data generated may contain some hints as to what signalling events may drive the LPM fate?

We agree that it would be useful to derive some hints regarding the signaling events that drive LPM fate. Indeed, our data provide suggestive evidence that hedgehog signaling may be involved. Three of the genes we found to be upregulated in CEP83-deficient cells are known target genes of hedgehog, namely OSR1, FOXF1, and FOXF2^1,2^. In addition, HAND2, which is strongly induced in CEP83-deficient cells, has been reported to be upstream of Hedgehog signaling^3^. Hedgehog plays an essential role in embryonic development and it is known to be a pathway regulated by the primary cilium^4-6^. Upon inspection of bulk-RNA seq data, we noted that two additional Hedgehog signaling components, GLI1 and PTCH1, were upregulated in CEP83^-/-^ cells^7-9^. We added a new Figure 6—figure supplement 3 to include these data. We also added to discuss a potential involvement of Hedgehog signaling: “In addition, CEP83-deficient cells downregulated expression of several transcripts encoding ciliary components (Figure 5—figure supplement 3) and evidence of an activation of several hedgehog signaling-associated genes. The primary cilium is critically involved in hedgehog signaling^10^. Moreover, Hedgehog signaling is important for mesodermal lineage decisions during gastrulation^11^. This raises the possibility that CEP83 controls mesodermal cell fate decisions by modulating hedgehog signaling in the mesoderm. Nevertheless, additional studies will be necessary to address this possibility.”

Could trajectories help in this regard or are there technical limitations that prohibit this?

We thank the reviewer for this interesting suggestion. We tried to utilize the single cell data to derive information regarding trajectories leading to LPM transition. However, it turned out that the primary driver of single cell transcriptome clusters was the cell cycle, and nascent LPM cells located in different clusters, precluding the option to carry out meaningful trajectory analyses.

And a major question that remains is if the phenotype is due to a ciliary defect or extraciliary function of CEP83.

We agree that this is an important question. So far our data are limited. We observed abnormal cilia formation in CEP83 mutant cells. We also observed decreased expression of mRNAs encoding components of the ciliary machinery in CEP83 mutant cells, a finding that coincides with ectopic LPM marker induction. This data was now added to the supplement (Figure 5—figure supplement 3). However, we do not know whether or not abnormal cilia formation causally contributes to the cell fate phenotype. We further added to the discussion to include this limitation.

It is interesting that Osr1 is upregulated in the CEP83-/- cells, but what about other more traditional nephron progenitor makers, such as SIX2 or CITED?

While SIX2 mRNA was not expressed at detectable levels, CITED1 was expressed in clusters 1, 3 and 4 (nephron progenitor) cells and down-regulated in CEP83 ^-/-^ cells compared to WT. We added a new supplementary figure (Figure 5—figure supplement 2) showing the expression of CITED1 in the three nephron progenitor clusters.

The percentage of contributing cells to the clusters is interesting, but can this data be boosted by statistics to evaluate if this is literally significant or not?

Since we only have two biological replicates of scRNA-seq data from KO and WT clones, we could not perform a formal statistical test on percentages of cells. We modified the wording of the Results section to clarify that the percentages were “numerically” lower in KO compared to WT cells.

I noticed that cluster 5 (damaged cells) seemed to be larger in the KO condition. What kind of damage does this entail and to what degree might this influence the broader interpretation of the experiment?

It is true that we observed more damaged cells among CEP83^-/-^ cells and we added additional data and discussion regarding this observation. The percentage of cluster 5 (damaged) cells is numerically higher in CEP83^-/-^ cells compared to WT cells at day 7. Cluster 5 cells express high levels of mitochondrial RNA, suggesting that they are injured or undergo apoptosis. We performed staining for active caspase 3 and found increased numbers of such apoptotic cells in CEP83^-/-^ cells compared to WT cells at day 7. These data were now added in the new Figure 4—figure supplement 3. One possible reason for increased apoptosis in CEP83^-/-^ cells is the depletion of HOXB7-positive ureteric bud progenitors, since loss of these cells is known to induce apoptosis of metanephric mesenchyme nephron progenitors in the intermediate mesoderm. We added this possibility to the Discussion section: “Furthermore, inductive signals from HOXB7-positive ureteric bud are known to maintain viability of nephron progenitor cells in the intermediate mesoderm^12^, which may contribute to increased numbers of apoptotic cells we observed in CEP83^-/-^ cells. “

1 Everson, J. L., Fink, D. M., Yoon, J. W., Leslie, E. J., Kietzman, H. W., Ansen-Wilson, L. J., Chung, H. M., Walterhouse, D. O., Marazita, M. L. & Lipinski, R. J. Sonic hedgehog regulation of Foxf2 promotes cranial neural crest mesenchyme proliferation and is disrupted in cleft lip morphogenesis. *Development (Cambridge, England)* 144, 2082-2091, doi:10.1242/dev.149930 (2017).

2 Han, L., Xu, J., Grigg, E., Slack, M., Chaturvedi, P., Jiang, R. & Zorn, A. M. Osr1 functions downstream of Hedgehog pathway to regulate foregut development. *Developmental Biology* 427, 72-83, doi:https://doi.org/10.1016/j.ydbio.2017.05.005 (2017).

3 Anderson, E., Peluso, S., Lettice, L. A. & Hill, R. E. Human limb abnormalities caused by disruption of hedgehog signaling. *Trends in Genetics* 28, 364-373 (2012).

4 Gazea, M., Tasouri, E., Tolve, M., Bosch, V., Kabanova, A., Gojak, C., Kurtulmus, B., Novikov, O., Spatz, J., Pereira, G., Hübner, W., Brodski, C., Tucker, K. L. & Blaess, S. Primary cilia are critical for Sonic hedgehog-mediated dopaminergic neurogenesis in the embryonic midbrain. *Developmental Biology* 409, 55-71, doi:https://doi.org/10.1016/j.ydbio.2015.10.033 (2016).

5 Pejskova, P., Reilly, M. L., Bino, L., Bernatik, O., Dolanska, L., Ganji, R. S., Zdrahal, Z., Benmerah, A. & Cajanek, L. KIF14 controls ciliogenesis via regulation of Aurora A and is important for Hedgehog signaling. *The Journal of cell biology* 219, doi:10.1083/jcb.201904107 (2020).

6 Takeda, S., Yonekawa, Y., Tanaka, Y., Okada, Y., Nonaka, S. & Hirokawa, N. Left-Right Asymmetry and Kinesin Superfamily Protein KIF3A: New Insights in Determination of Laterality and Mesoderm Induction by kif3A−/− Mice Analysis. *Journal of Cell Biology* 145, 825-836, doi:10.1083/jcb.145.4.825 (1999).

7 Litingtung, Y., Lei, L., Westphal, H. & Chiang, C. Sonic hedgehog is essential to foregut development. *Nature genetics* 20, 58-61 (1998).

8 Lee, J., Platt, K. A., Censullo, P. & Ruiz i Altaba, A. Gli1 is a target of Sonic hedgehog that induces ventral neural tube development. *Development (Cambridge, England)* 124, 2537-2552 (1997).

9 Villavicencio, E. H., Walterhouse, D. O. & Iannaccone, P. M. The sonic hedgehog–patched–gli pathway in human development and disease. *American journal of human genetics* 67, 1047 (2000).

10 Ho, E. K. & Stearns, T. Hedgehog signaling and the primary cilium: implications for spatial and temporal constraints on signaling. *Development (Cambridge, England)* 148, doi:10.1242/dev.195552 (2021).

11 Guzzetta, A., Koska, M., Rowton, M., Sullivan, K. R., Jacobs-Li, J., Kweon, J., Hidalgo, H., Eckart, H., Hoffmann, A. D., Back, R., Lozano, S., Moon, A. M., Basu, A., Bressan, M., Pott, S. & Moskowitz, I. P. Hedgehog-FGF signaling axis patterns anterior mesoderm during gastrulation. *Proceedings of the National Academy of Sciences of the United States of America* 117, 15712-15723, doi:10.1073/pnas.1914167117 (2020).

12 Barasch, J., Qiao, J., McWilliams, G., Chen, D., Oliver, J. A. & Herzlinger, D. Ureteric bud cells secrete multiple factors, including bFGF, which rescue renal progenitors from apoptosis. *The American journal of physiology*
**273**, F757-767, doi:10.1152/ajprenal.1997.273.5.F757 (1997).